# Bacteria are a major determinant of Orsay virus transmission and infection in *Caenorhabditis elegans*

**Brian G Vassallo[1,2], Noemie Scheidel[1], Sylvia E J Fischer[1], Dennis H Kim[1]***

[1]Division of Infectious Diseases, Department of Pediatrics, Boston Children's Hospital and Harvard Medical School, Boston, United States; [2]Department of Biology, Massachusetts Institute of Technology, Cambridge, United States

*For correspondence: dennis.kim@childrens.harvard.edu

**Competing interest:** The authors declare that no competing interests exist.

**Abstract** The microbiota is a key determinant of the physiology and immunity of animal hosts. The factors governing the transmissibility of viruses between susceptible hosts are incompletely understood. Bacteria serve as food for *Caenorhabditis elegans* and represent an integral part of the natural environment of *C. elegans*. We determined the effects of bacteria isolated with *C. elegans* from its natural environment on the transmission of Orsay virus in *C. elegans* using quantitative virus transmission and host susceptibility assays. We observed that *Ochrobactrum* species promoted Orsay virus transmission, whereas *Pseudomonas lurida* MYb11 attenuated virus transmission relative to the standard laboratory bacterial food *Escherichia coli* OP50. We found that pathogenic *Pseudomonas aeruginosa* strains PA01 and PA14 further attenuated virus transmission. We determined that the amount of Orsay virus required to infect 50% of a *C. elegans* population on *P. lurida* MYb11 compared with *Ochrobactrum vermis* MYb71 was dramatically increased, over three orders of magnitude. Host susceptibility was attenuated even further in the presence of *P. aeruginosa* PA14. Genetic analysis of the determinants of *P. aeruginosa* required for attenuation of *C. elegans* susceptibility to Orsay virus infection revealed a role for regulators of quorum sensing. Our data suggest that distinct constituents of the *C. elegans* microbiota and potential pathogens can have widely divergent effects on Orsay virus transmission, such that associated bacteria can effectively determine host susceptibility versus resistance to viral infection. Our study provides quantitative evidence for a critical role for tripartite host-virus-bacteria interactions in determining the transmissibility of viruses among susceptible hosts.

## eLife assessment

Using a *C. elegans*/virus system, this **important** work demonstrates that viral susceptibility can be greatly altered by the bacterial food that *C. elegans* consumes. The work is rigorous with **solid** support for the conclusions: the authors show that quorum-sensing compounds play a role in reducing host susceptibility, and they perform control experiments to rule out nutrition and pathogenicity of the bacteria as the cause of impacts on viral susceptibility.

## Introduction

Viruses are ubiquitous and abundant (*Edwards and Rohwer, 2005*; *Srinivasiah et al., 2008*). Infection can have profound consequences for the health of an individual host. The ability of viruses to transmit from one individual to another can scale these consequences causing morbidity and mortality throughout whole populations. Many factors influence virus transmission (*Pica and Bouvier, 2012*; *Thangavel and Bouvier, 2014*; *de Vries et al., 2021*). Abiotic factors such as temperature (*Lowen*

*et al., 2007*) and humidity (*Schulman and Kilbourne, 1962*) and biotic factors such as viral load (*Quinn et al., 2000*; *Edenborough et al., 2012*) and host immune status (*Price et al., 2014*; *Chua et al., 2015*) all interact to determine transmission rates. Despite these findings, the determinants of virus transmissibility remain incompletely understood.

The microbiota has emerged as a host-associated factor that modulates multiple aspects of virus infection and thereby alters transmission rates among host organisms (*Kane et al., 2011*; *Kuss et al., 2011*; *Wu et al., 2019*). In general, the microbiota is critical for the proper development of the immune system and for the effective activation of antimicrobial immune responses even at sites distal to microbiota colonization (*Zheng et al., 2020*). More specifically, bacteria and bacterial surface structures such as lipopolysaccharide and peptidoglycan have been shown to stabilize poliovirus and reovirus in vitro (*Kuss et al., 2011*; *Berger et al., 2017*). These observations likely explain why microbiota depletion by antibiotic treatment was sufficient to provide protection against the same viruses in mice (*Kuss et al., 2011*). The bacterial symbiont *Wolbachia* protects numerous insect species from multiple viruses either by upregulating antiviral defenses or competing for intracellular nutrients (*Hedges et al., 1979*; *Teixeira et al., 2008*; *Moreira et al., 2009*; *Bian et al., 2010*; *Pimentel et al., 2020*). Individual bacteria have also been found to enhance viral infection; *Serratia marcescens* promoted infection of the mosquito *Aedes aegypti* by Dengue, Zika, and Sindbis viruses by secreting a protein, enhancin, that degrades the mucus layer covering epithelial cells (*Wu et al., 2019*).

*Caenorhabditis elegans* is a nematode often found in microbially rich environments such as rotting vegetation (*Schulenburg and Félix, 2017*). The first naturally occurring virus capable of infecting the *C. elegans* was isolated from Orsay, France (*Félix et al., 2011*). Orsay virus is a part of a group of nematode infecting viruses closely related to the Nodaviridae family of viruses which infect arthropods and fish (*Félix et al., 2011*; *Félix and Wang, 2019*). Like other Nodaviruses, Orsay virus is a positive-sense, single-stranded RNA virus with a bipartite genome (*Félix et al., 2011*). Transmission of Orsay virus occurs horizontally through the fecal-oral route. Fluorescence in situ hybridization and immunofluorescence imaging has revealed that Orsay virus solely infects *C. elegans* intestinal cells (*Félix et al., 2011*; *Franz et al., 2014*). Viral infection activates host defense mechanisms including the RNA-interference response and a transcriptional program known as the intracellular pathogen response (*Félix et al., 2011*; *Sarkies et al., 2013*; *Bakowski et al., 2014*; *Chen et al., 2017*; *Le Pen et al., 2018*).

*C. elegans* is a bacterivore and is propagated in the laboratory on lawns of *Escherichia coli* OP50. *Pseudomonas aeruginosa,* an opportunistic pathogen of humans, is found in the soil and water and can also infect *C. elegans* (*Tan et al., 1999a*; *Crone et al., 2020*; *Pelegrin et al., 2021*). Infection of *C. elegans* with *P. aeruginosa* activates innate immunity and stress responses, as well as behavioral avoidance responses (*Kim et al., 2002*; *Kim and Ewbank, 2018*; *Zhang et al., 2005*). Recently, the bacteria resident in the natural environment of *C. elegans* in the wild have been of increasing interest (*Schulenburg and Félix, 2017*; *Dirksen et al., 2016*; *Samuel et al., 2016*; *Berg et al., 2016*). The intestinal lumen of free-dwelling *C. elegans* is occupied by taxonomically and functionally diverse bacteria that can affect *C. elegans* fitness and physiology (*Dirksen et al., 2016*; *Samuel et al., 2016*; *Dirksen, 2020*).

In this study, we sought to understand how bacteria that are constituents of the *C. elegans* microbiota quantitatively affect the transmission of Orsay virus in *C. elegans*. We observed that monoaxenic cultures of different bacterial species had widely divergent effects on the transmission of Orsay virus and conducted genetic analysis of the bacterial determinants involved in modulating virus transmission. Our data point to a key species-specific role for bacteria as critical determinant of virus transmission.

## Results

### Wide variation in the effects of bacteria on the transmission of Orsay virus

Orsay virus spreads horizontally through the fecal-oral route and transmission can spread from a single animal to a population of animals on a plate (*Yuan et al., 2018*). We sought to assess the impact of bacteria on Orsay virus transmission rates and utilized a collection of bacteria isolated from the environment with wild *C. elegans* to assemble a panel of Gram-negative bacteria for comparison

with the standard laboratory bacterial food, *E. coli* OP50 (*Dirksen et al., 2016*; *Samuel et al., 2016*; *Dirksen, 2020*; *Brenner, 1974*; *Troemel et al., 2008*). All animals were raised to young adulthood on lawns of *E. coli* OP50 to eliminate differences in developmental rates caused by different bacteria (*Dirksen, 2020*). We set up a transmission assay by placing infected animals ('spreaders') together with uninfected animals on a monoaxenic lawn of each test bacterium. We quantified the incidence proportion, defined as the number of new infections produced after 24 hr as determined by detection of induction of the *pals-5p::GFP* reporter, which is induced by infection with Orsay virus (*Figure 1A–C*; *Bakowski et al., 2014*). We observed a wide range in the measured incidence proportion (*Figure 1B and C*). Exposure to many of the naturally associated bacterial strains resulted in transmission that was comparable to the incidence proportion observed with *E. coli* OP50 with some prominent exceptions (*Figure 1B and C*). Two *Ochrobactrum* species promoted virus infection in nearly all individuals in the transmission assay and increased the incidence proportion 2.7-fold and 2.9-fold compared to *E. coli* OP50 (*Figure 1B and C*). We confirmed that transmission from the initial spreader animals in the assay, and not multiple rounds of infection, was responsible for the increased incidence proportion observed in the presence of *Ochrobactrum vermis* MYb71 (*Figure 1—figure supplement 1*). On the other hand, the presence of *Pseudomonas lurida* MYb11 reduced the incidence proportion 4.1-fold compared to *E. coli* OP50 and 11-fold compared to *O. vermis* MYb71 (*Figure 1B and C*). These results present a striking divergence in the effects of individual bacterial constituents of the *C. elegans* microbiota on the level of transmission of Orsay virus from infected animals.

We reasoned that increased transmission of Orsay virus could be due to bacterial modulation of host susceptibility to viral infection. To quantify the degree of host susceptibility to Orsay virus we added Orsay virus in doses that ranged across four orders of magnitude and then measured the fraction of individuals that became *pals-5p::GFP* positive after 24 hr (*Figure 1A and D*, *Figure 1—figure supplements 2 and 3*). We also specifically calculated the $ID_{50}$, or the dose of Orsay virus required to infect 50% of the population after 24 hr of exposure (*Figure 1D and E* and *Figure 1—figure supplement 2*). On average, 3.6 µL of Orsay virus stock prepared as indicated in the Materials and methods was required to infect 50% of a population of ZD2611 animals in the presence of *E. coli* OP50 after 24 hr (*Figure 1—figure supplement 2*). We observed variability in the measured $ID_{50}$ even when using the same batch of Orsay virus (e.g. *Figure 1D* vs *Figure 1—figure supplement 3A* vs. *Figure 1—figure supplement 3B*). We therefore chose to control all our experiments internally rather than attempt to normalize between Orsay virus batches and we report doses used in arbitrary units (a.u.), however 1 a.u. corresponds to 1 µL of Orsay virus filtrate (Materials and methods).

To determine quantitatively how different bacterial species modulate host susceptibility to viral infection independently of the potential differential effects of bacteria on host shedding of viruses, we added fixed doses of Orsay virus to plates of each bacterial species and monitored infection using the *pals-5p::GFP* reporter. At lower doses of Orsay virus, the presence of *O. vermis* MYb71 resulted in a higher fraction of infected animals than those observed in the presence of *E. coli* OP50 or *P. lurida* MYb11 (*Figure 1D*, *Figure 1—figure supplement 3*). At higher doses of Orsay virus, the presence of *P. lurida* MYb11 resulted in a reduced fraction of infected animals compared to *O. vermis* MYb71 and *E. coli* OP50 (*Figure 1D*, *Figure 1—figure supplement 3*). We further quantified the $ID_{50}$ in the presence of each bacterium and on average we observed that the $ID_{50}$ in the presence of *E. coli OP50* was 14-fold higher than the $ID_{50}$ in the presence of *O. vermis* MYb71 (*Figure 1D and E*, and *Figure 1—figure supplement 3*). The $ID_{50}$ observed in the presence of *P. lurida* was 120-fold and 1800-fold higher than the $ID_{50}$ in the presence of *E. coli* OP50 or *O. vermis* MYb71, respectively (*Figure 1D and E*, and *Figure 1—figure supplement 3*).

We corroborated our observations from the *pals-5p::GFP* reporter by scoring the same samples using fluorescence in situ hybridization to detect the RNA1 segment of the Orsay virus genome in the intestinal cells of infected animals (*Figure 1F*). As expected, we observed that at lower doses the presence of *O. vermis* MYb71 resulted in a greater fraction of infected animals compared to *E. coli* OP50 or *P. lurida* MYb11, while at higher doses the presence of *P. lurida* MYb11 resulted in a reduced fraction of infected animals (*Figure 1F*). Together these data establish that individual members of the *C. elegans* microbiota can modulate host susceptibility to Orsay virus over three orders of magnitude with dramatic consequences for the transmissibility of Orsay virus.

We also assessed whether bacteria altered host feeding behavior as this may influence the transmission and infection rate of Orsay virus. We quantified the number of pharyngeal pumps per minute

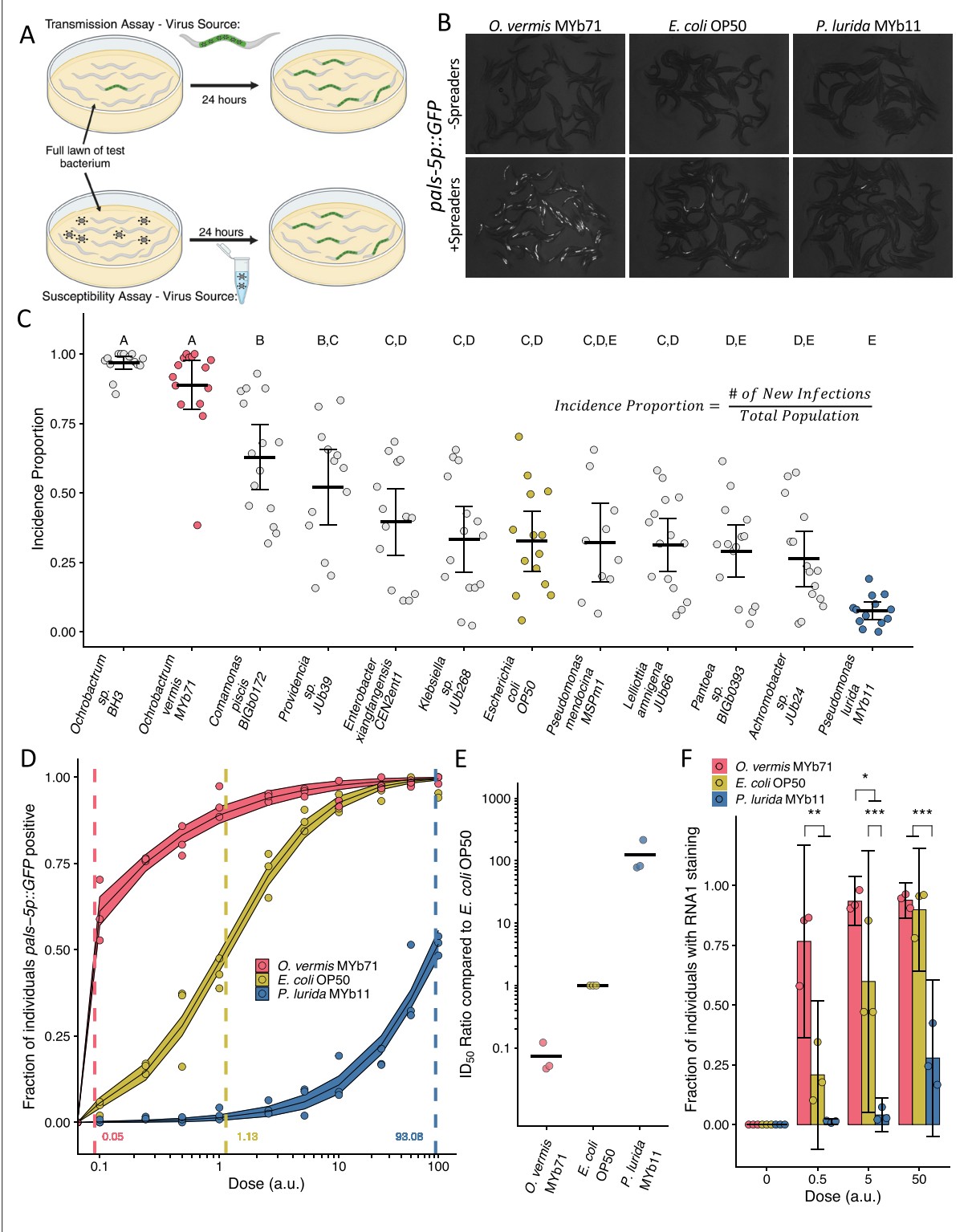

$$Incidence\ Proportion = \frac{\#\ of\ New\ Infections}{Total\ Population}$$

**Figure 1.** *Ochrobactrum* species and *P. lurida* MYb11 divergently modulate Orsay virus transmission and infection rates. (**A**) Schematic representation of the transmission and susceptibility assays: Transmission can be assessed by combining infected spreader individuals (green nematode), uninfected reporter individuals (white nematodes), and various bacteria. Susceptibility can be assessed by combining uninfected reporter individuals with exogenous Orsay virus, and various bacteria. 24 hr later, infection is assessed. (**B**) Representative images of *pals-5p::GFP* expression among individuals exposed to spreaders or no spreaders in the presence of *O. vermis* MYb71, *E. coli* OP50, or *P. lurida* MYb11. (**C**) Incidence proportion, calculated as indicated, of Orsay virus transmission quantified on different bacteria from the environment of *C. elegans*. Data shown are from three experiments

*Figure 1 continued on next page*

*Figure 1 continued*

combined, each dot represents the incidence proportion from a single plate, and letters denote statistical significance. Treatments with the same letter are not significantly different from one another. (n=19,694 in total and n>42 for all dots). (**D**) Dose-response curves of *C. elegans* to exogenous Orsay virus. The dashed line indicates the calculated dose at which 50% of the population was infected, the ID50, the exact value which is given above the x-axis for each bacterium. The solid curves represent the 95% confidence interval of the modeled log-logistic function while in the presence of each bacterium. Data are from a single representative experiment and replicates can be found in *Figure 1—figure supplement 3*. Dots represent individual plates. (a.u., arbitrary units, see Materials and methods) (n=8334 in total and n>32 for all dots). (**E**) Ratios of the ID50 calculated on the indicated bacteria as compared to the ID50 calculated on *E. coli* OP50 from three separate experiments as in (**D**) and *Figure 1—figure supplement 3*. (**F**) The fraction of individuals with staining as assessed by fluorescence in situ hybridization targeting the RNA1 segment of Orsay virus. Data shown are from three experiments combined, each dot represents three pooled technical replicate plates from the susceptibility assays in (**D**) and *Figure 1—figure supplement 3* (n=3446 in total and n>61 for all dots). For all plots the black bar is the mean and error bars are the 95% confidence interval (C.I.). p-Values determined using one-way ANOVA followed by Tukey's honest significant difference (HSD) test (NS, non-significant, *p<0.05, **p<0.01, ***p<0.001).

The online version of this article includes the following figure supplement(s) for figure 1:

**Figure supplement 1.** Transmission does not occur from individuals infected during the course of a transmission or susceptibility assay.

**Figure supplement 2.** The range of all ID50 values calculated for each batch of Orsay virus.

**Figure supplement 3.** Replicate dose-response curves shown in *Figure 1D*.

**Figure supplement 4.** Pharyngeal pumping rates are unaltered at 6 hr and change only slightly by 24 hr after exposure to novel bacteria.

(ppm), a metric of *C. elegans* feeding, in the presence of each bacterium. After 6 hr of exposure to *E. coli* OP50, *P. lurida* MYb11, or *O. vermis* MYb71, there were no differences in pharyngeal pumping rates (*Figure 1—figure supplement 4A*). At 24 hr pumping rates were decreased in the presence of *O. vermis* MYb71 to an average of 242 ppm compared to 261 ppm in the presence of *P. lurida* MYb11 suggesting that changes to feeding behavior are unlikely to be responsible for observed effects on host susceptibility to Orsay virus (*Figure 1—figure supplement 4B*).

## *P. aeruginosa* attenuates Orsay virus transmission

In view of the effect of *P. lurida* MYb11 on attenuating Orsay virus infection of *C. elegans,* we examined the effect of the distantly related bacterium, *Pseudomonas aeruginosa,* an opportunistic pathogen of humans that has been characterized extensively (*Tan et al., 1999a*; *Mahajan-Miklos et al., 1999*). We observed that in the presence of either *P. aeruginosa* strains PA01 or PA14, transmission from spreader animals to uninfected individuals was nearly completely blocked and the incidence proportion was 11-fold and 32-fold lower than that observed in the presence of *E. coli* OP50, respectively (*Figure 2A*). The attenuating effect of *P. aeruginosa* PA01 and PA14 on virus transmission was further reduced 3.1-fold and 9.5-fold, respectively, compared to the incidence proportion observed in the presence of *P. lurida* MYb11 (*Figure 2A*).

We confirmed that host susceptibility to Orsay virus was reduced in the presence of *P. aeruginosa* by performing susceptibility assays at two doses of exogenous Orsay virus (*Figure 2B*, *Figure 2—figure supplement 1*). At each dose, fewer animals were infected following exposure to equivalent doses of Orsay virus in the presence of *P. aeruginosa* PA01, PA14, or *P. lurida* MYb11 as compared to the fraction of animals infected in the presence of *E. coli* OP50 (*Figure 2B*, *Figure 2—figure supplement 1*). At the highest dose of virus used, we still observed robust attenuation of infection in the presence of *P. aeruginosa* PA14 as 0.5% of the animals were infected as compared to 54%, 70%, and 97% in the presence of *P. aeruginosa* PA01, *P. lurida* MYb11, or *E. coli* OP50, respectively (*Figure 2B*). We confirmed that no individuals were detectably infected with Orsay virus while in the presence of *P. aeruginosa* PA14 using FISH staining (*Figure 2C*). FISH additionally confirmed that the presence of *P. aeruginosa* PA01 and *P. lurida* MYb11 attenuated average infection to 24% and 60% of the population respectively compared to *E. coli* OP50 which supported infection of 86% of the population. Further, *P. aeruginosa* PA01, *P. lurida* MYb11, and *E. coli* OP50 all supported higher levels of infection than *P. aeruginosa* PA14 (*Figure 2C*). Together these results demonstrate a striking capacity of *P. lurida* MYb11 and *P. aeruginosa* strains to sharply reduce, and even effectively block, host susceptibility to infection with Orsay virus.

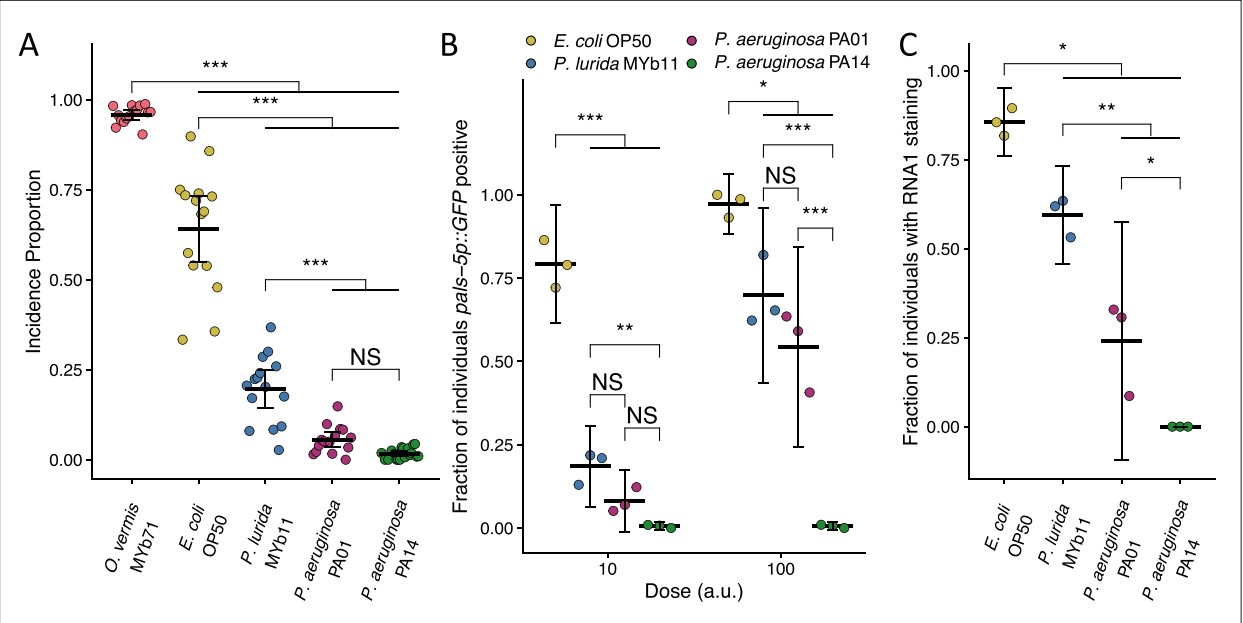

**Figure 2.** *P. aeruginosa* attenuates Orsay virus transmission and infection rates. (**A**) Incidence proportion of Orsay virus transmission quantified on different bacteria from *C. elegans*' environment and *P. aeruginosa* PA01 and *P. aeruginosa* PA14. Data shown are from three experiments combined, each dot represents the incidence proportion from a single plate (n=9214 in total and n>54 for all dots). (**B**) The fraction of individuals that became *pals-5p::GFP* positive following exposure to two doses of exogenous Orsay virus. Data are from a single representative experiment and replicates can be found in *Figure 2—figure supplement 1*. Dots represent individual plates (n=6539 in total and n>56 for all dots). (**C**) The fraction of individuals with staining following exposure to 100 a.u. Orsay virus as assessed by fluorescence in situ hybridization targeting the RNA1 segment of the Orsay virus genome. Data shown are from three experiments combined, each dot represents three pooled technical replicate plates from (**B**) and *Figure 2— figure supplement 1* (n=2743 in total and n>116 for all dots). For all plots the black bar is the mean and error bars are the 95% confidence interval (C.I.). p-Values determined using one-way ANOVA followed by Tukey's honest significant difference (HSD) test (NS, non-significant, *p<0.05, **p<0.01, ***p<0.001) (a.u., arbitrary units).

The online version of this article includes the following figure supplement(s) for figure 2:

**Figure supplement 1.** Replicate susceptibility assays shown in *Figure 2B*.

**Figure supplement 2.** *O. vermis* MYb71, *P. aeruginosa* PA14, *P. lurida* MYb11, and *P. aeruginosa* PA01 reduce lifespan compared to *E. coli* OP50.

We again assessed whether bacterial modulation of host feeding behavior may be responsible for the observed effects on host susceptibility to Orsay virus. There were no differences observed in pharyngeal pumping rate after 6 hr of exposure to each bacterium (*Figure 1—figure supplement 4A*). After 24 hr pharyngeal pumping rate increased in the presence of *P. aeruginosa* PA14 to an average of 264 ppm compared to 248 ppm in the presence of *E. coli* OP50 which suggests that changes to feeding behavior are unlikely to account for the attenuation of infection observed in *P. aeruginosa* (*Figure 1—figure supplement 4B*).

*P. aeruginosa* is a pathogen of *C. elegans* and numerous other organisms (*Tan et al., 1999a*; *Rahme et al., 1995*; *Tan et al., 1999b*). Moreover, while both PA01 and PA14 are pathogenic, PA14 is more virulent toward *C. elegans* (*Tan et al., 1999a*). *P. lurida* MYb11 promotes developmental rate and fitness, although does so at a minor cost to overall lifespan (*Dirksen, 2020*; *Kissoyan et al., 2022*). The effect of *O. vermis* MYb71 on lifespan has not been assessed. We assessed host lifespan under our assay conditions to examine links between bacterial pathogenicity and host susceptibility to Orsay virus. Under our assay conditions, including a full lawn of bacteria, which prevent *C. elegans* from avoiding undesirable bacteria, and a temperature of 20°C, which is below the optimal virulence temperature of *P. aeruginosa*, substantial mortality is not observed until 72 hr after exposure (*Figure 2—figure supplement 2*). All bacteria enhanced mortality in comparison to *E. coli* OP50 (*Figure 2—figure supplement 2*). Interestingly, *O. vermis* MYb71 exposure led to similarly enhanced mortality when compared to *P. aeruginosa* PA14, and both bacteria enhanced mortality in comparison to *P. lurida* MYb11 or *P. aeruginosa* PA01 (*Figure 2—figure supplement 2*). *P. lurida* MYb11 also enhanced mortality compared to *P. aeruginosa* PA01 under these conditions (*Figure 2—figure*

*supplement 2*). The similar lifespans of animals exposed to *P. aeruginosa* PA14 and *O. vermis* MYb71 stand in contrast to their divergent effects on host susceptibility to Orsay virus. This suggests that general pathogenicity is unlikely to be responsible for effects on host susceptibility, although pathogenic consequences specific to exposure to each bacterium may still contribute to the observed effects on host susceptibility to Orsay virus.

## Attenuation of Orsay virus transmission by *P. lurida* and *P. aeruginosa* is not due to effects on Orsay virus replication, stability, or shedding

We considered the possibility that the attenuation of Orsay virus transmission in the presence of *P. lurida* and *P. aeruginosa* strains might be due to inhibitory effects of the bacteria on the replication of Orsay virus once transmitted to a susceptible animal host. To evaluate this possibility, we made use of a plasmid-based system in which viral RNA1 is expressed through a transgene introduced into *C. elegans,* so that replication of RNA1 can be assessed independent of the entry of exogenous virus into the host (*Jiang et al., 2017*). In this system, the Orsay virus RNA1 segment, which encodes the RNA-dependent RNA polymerase (RdRP), is expressed following heat-shock. The expressed RNA1 may then be translated to produce the RdRP which can then replicate RNA1 through a negative-strand intermediate. The expression of RNA1 via heat-shock bypasses any differences in viral entry or pre-replication steps, allowing for a direct test of RNA1 replication efficiency under different conditions (*Jiang et al., 2017*). A plasmid expressing a mutated RdRP (RNA1[D601A]) incapable of supporting further RNA1 replication after the initial heat-shock serves as a control for heat-shock efficiency (*Jiang et al., 2017*). Using this system, we observed that Orsay virus RNA1 replication efficiency was unaffected by the presence of *Pseudomonas* species relative to RNA1 replication observed in the presence of *E. coli* OP50 (*Figure 3A*). Additionally, there were no differences observed in heat-shock efficiency between the different bacteria (*Figure 3A*). These data suggest that bacteria-induced differences in RNA1 replication in the host do not explain the substantial attenuation of Orsay virus transmission and infection rates caused by *P. aeruginosa* PA01 or PA14 and *P. lurida* MYb11.

We sought to confirm the successful replication of the plasmid expressed RNA1 of Orsay virus by assessing whether we could detect the replication of Orsay virus RNA1 in each *Pseudomonas* species. We exposed young adult wild-type (N2) or RNAi defective animals (*rde-1(ne219)*) to exogenous Orsay virus and quantified the amount of virus present at 2 hr and 24 hr post exposure, with the difference in RNA levels at these points reflecting viral genome replication. Animals of the RNAi defective *rde-1(ne219)* background have a defective antiviral response leading to higher viral loads than wild-type animals (*Félix et al., 2011*). The *rde-1(ne219)* mutant therefore provides a more sensitive genetic background in which to detect viral replication following rare infection events within a population as is expected in the presence of *P. aeruginosa* PA14. In *rde-1(ne219)* animals viral RNA1 levels increased at 24 hr relative to 2 hr post exposure regardless of the bacteria present (*Figure 3B and C*). On the other hand, in the wild-type background replication was not observed in animals exposed to *P. aeruginosa* PA14 and was attenuated in animals exposed to *P. aeruginosa* PA01 compared to animals exposed to *E. coli* OP50 or *P. lurida* MYb11 (*Figure 3B and C*). In a wild-type background individuals exposed to *P. aeruginosa* PA01 and *P. aeruginosa* PA14 are less likely to become infected from exogenous Orsay virus (*Figure 2B*, *Figure 2—figure supplement 1*). It is therefore likely that the viral load we observed in the wild-type populations is due to a smaller proportion of the population being infected, rather than hampered replication (*Félix et al., 2011*).

We also assessed whether *P. lurida* MYb11, *P. aeruginosa* PA01, or *P. aeruginosa* PA14 was capable of rapidly degrading Orsay virus or preventing viral shedding, either of which might account for attenuation of Orsay virus transmission. We did not observe a substantial decrease in viral stability in the presence of any bacterium suggesting Orsay virus is not rapidly degraded (*Figure 3D and E*). We did observe an approximately twofold decrease in viral levels between *E. coli* OP50 and *P. lurida* MYb11, however, this difference already existed within 30 min of exposure to each bacterium and did not grow further over the course of 24 hr suggesting we are measuring differences in recovery of Orsay virus from each bacterial lawn, rather than enhanced degradation in the presence of *P. lurida* MYb11 (*Figure 3D and E*). We measured rates of Orsay virus shedding during a transmission assay. We did not observe a substantial decrease in viral shedding in the presence of any bacterium (*Figure 3F*). We observed approximately fourfold lower levels of Orsay virus shed in the presence of *P. lurida* MYb11 vs *E. coli* OP50, but as suggested by our virus stability data, we believe these differences at least partly

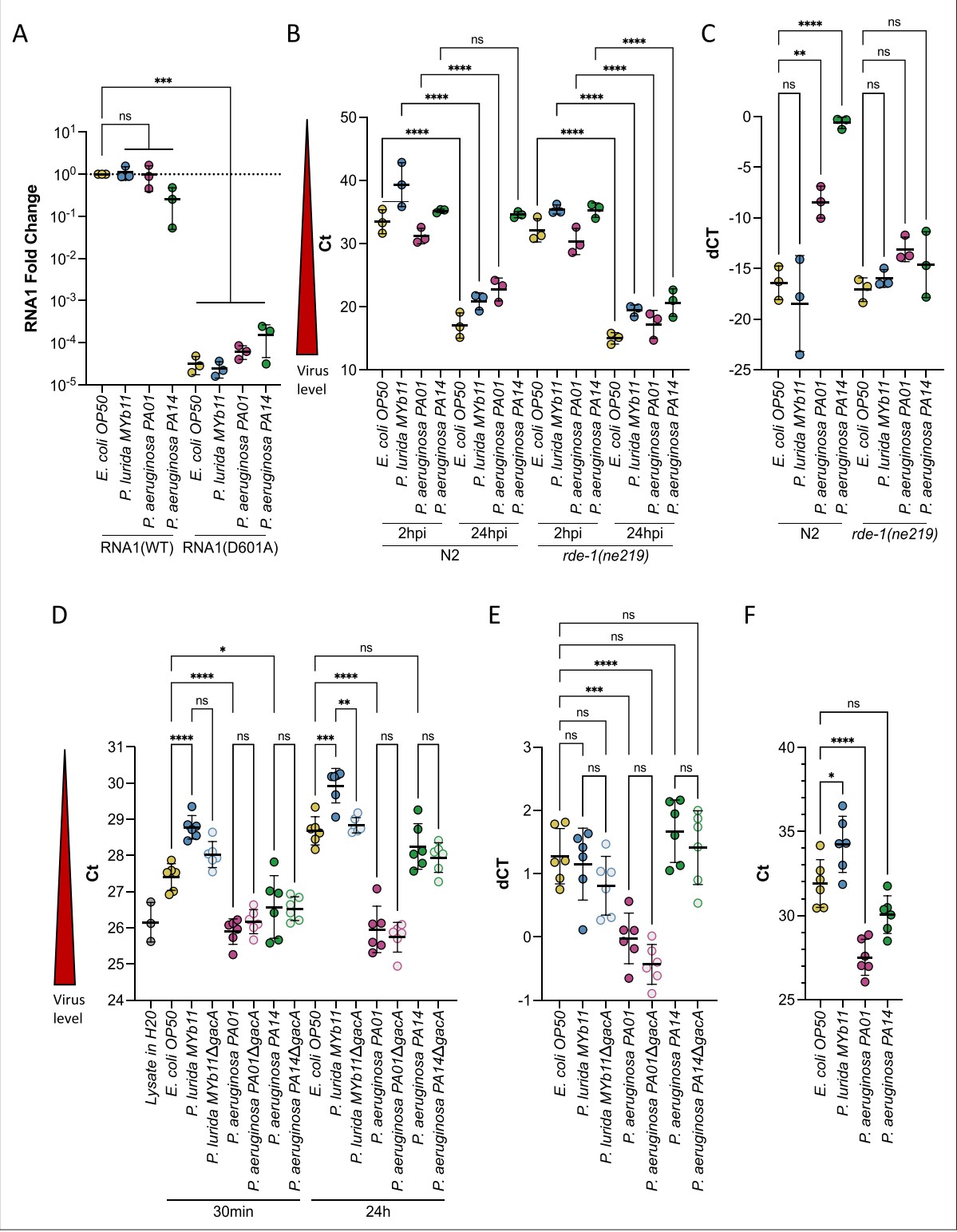

**Figure 3.** *P. aeruginosa* and *P. lurida* MYb11 do not eliminate Orsay virus replication, rapidly degrade Orsay virus, or eliminate Orsay virus shedding by infected spreader animals. (**A**) Animals carrying either a wild-type (RNA1(WT)) or replication defective (RNA1(D601A)) transgenic viral RNA replicon system (*Jiang et al., 2017*) were used to assess Orsay virus replication efficiency independently of virus entry. p-Values determined using one-way ANOVA followed by Dunnett's test. (**B**) Orsay virus RNA1 levels of N2 and *rde-1(ne219)* animals exposed to the indicated bacterium and exogenous Orsay virus 2 hr and 24 hr post infection (hpi). p-Values were determined using Welch's t-test. For each plot, each dot represents five pooled technical replicates, the black bar is the mean, error bars are the standard deviation, RNA1 levels were quantified by qPCR, and the data shown are for three

*Figure 3 continued on next page*

*Figure 3 continued*

independent experiments. (**C**) Delta Ct comparing the 24 hr and 2 hr timepoints from (**B**). (**D**) Stability of Orsay virus in the presence of lawns of the indicated bacteria for the indicated time at 20°C. (**E**) Delta Ct comparing the 24 hr and 30 min timepoints from (**D**). (**F**) Orsay virus RNA1 levels recovered from the indicated lawns after infected ZD2610(*rde- 1(ne219);jyIs8[pals-5p::GFP;myo-2p::mCherry];glp-4(bn2ts)*) spreaders were allowed to shed for 24 hr mimicking a transmission assay (NS, non-significant, *p<0.05, **p<0.01, ***p<0.001).

reflect differences in Orsay virus recovery from each bacterial lawn rather than actual differences in viral shedding (*Figure 3F*).

## Attenuation of Orsay virus transmission by *Pseudomonas* species is dependent on regulators of bacterial quorum sensing

Extensive studies on *Pseudomonas* species have demonstrated the importance of quorum sensing for regulating many community-level behaviors in response to growing population density (*Lee and Zhang, 2015*). *P. aeruginosa* relies upon three quorum sensing systems: *las*, *rhl*, and *pqs*. These systems are arranged hierarchically, however crosstalk between them is extensive (*Figure 4A*; *Lee and Zhang, 2015*). In *P. aeruginosa*, an additional layer of regulation stems from two-component regulatory systems such as *gacA/gacS* which regulates numerous genes that together influence quorum sensing, virulence, and biofilm development (*Figure 4A*; *Reimmann et al., 1997*; *Parkins et al., 2001*; *Brencic et al., 2009*). *P. aeruginosa* possesses three exopolysaccharide biosynthetic clusters that each contribute to biofilm formation: *pel*, *psl*, and *alg*. However, *P. aeruginosa* PA01 preferentially produces Psl while *P. aeruginosa* PA14 is unable to synthesize Psl and produce Pel (*Friedman and Kolter, 2004a*; *Friedman and Kolter, 2004b*; *Franklin et al., 2011*; *Jackson et al., 2004*; *Matsukawa and Greenberg, 2004*). We hypothesized that quorum sensing might mediate the effect of *Pseudomonas* strains to attenuate virus transmission, while differences in exopolysaccharide production might mediate the enhanced attenuation of virus transmission observed in the presence of *P. aeruginosa* PA14 compared with what was observed in the presence of *P. aeruginosa* PA01. Therefore, we tested a panel of *P. aeruginosa* PA01 and *P. aeruginosa* PA14 quorum sensing and biofilm mutants to determine whether quorum sensing or biofilm formation was involved in the attenuation of Orsay virus infection mediated by *P. aeruginosa*.

Mutations of any the regulators of the *las*, *rhl*, or *pqs* quorum sensing systems suppressed the attenuation of Orsay virus infection caused by the presence of wild-type *P. aeruginosa* PA01 (*Figure 4B*, *Figure 4—figure supplement 1*). Knockout of *gacA* in *P. aeruginosa* PA01 also suppressed the attenuation of Orsay virus infection (*Figure 4B*, *Figure 4—figure supplement 1*). On the other hand, mutation of any of the three exopolysaccharide production pathways had no potent effect on the attenuation of infection (*Figure 4B*, *Figure 4—figure supplement 1*). These data support a role for quorum sensing regulated processes in reducing Orsay virus infection rates but do not implicate a role for individual exopolysaccharides.

For *P. aeruginosa* PA14, mutation of *gacA* or *rhlR* suppressed the attenuation of Orsay virus infection observed in the presence of wild-type *P. aeruginosa* PA14 (*Figure 4C*, *Figure 4—figure supplement 2*). Loss of the *rhlR* regulators *rhlI* or *pqsE* alone had no effect on the attenuation of Orsay virus infection, but simultaneous mutation of both *rhlI* and *pqsE* did suppress the attenuation of Orsay virus infection observed in the presence of wild-type *P. aeruginosa* PA14, similar to that observed for the *rhlR* mutant, suggesting *rhlI* and *pqsE* function redundantly to regulate *rhlR* in this context (*Figure 4C*, *Figure 4—figure supplement 2*; *Mukherjee et al., 2018*). Mutation of *lasI* or *lasR* suppressed the attenuation of Orsay virus infection to a lesser extent (*Figure 4C*, *Figure 4—figure supplement 2*). Independent mutation of two genes responsible for *pel* or *alg* exopolysaccharide production had no effect on infection rates (*Figure 4C*, *Figure 4—figure supplement 2*). We further confirmed that mutation of *rhlR* and *gacA* in *P. aeruginosa* PA01 or *P. aeruginosa* PA14 suppressed the attenuation of Orsay virus infection via FISH (*Figure 4D and E*). These data suggest a role for quorum sensing in mediating *P. aeruginosa* PA14 suppression of Orsay virus infection, as we observed for *P. aeruginosa* PA01. However, our results obtained in the presence of *P. aeruginosa* PA14 suggest that there may be some differential regulation of the bacterial effectors responsible in comparison to *P. aeruginosa* PA01, or additional non-quorum sensing-related factors that also mediate suppression.

As *P. aeruginosa* mutants could suppress the attenuation of infection observed by wild-type *P. aeruginosa* in the presence of exogenous virus, we next confirmed that these *P. aeruginosa* mutants

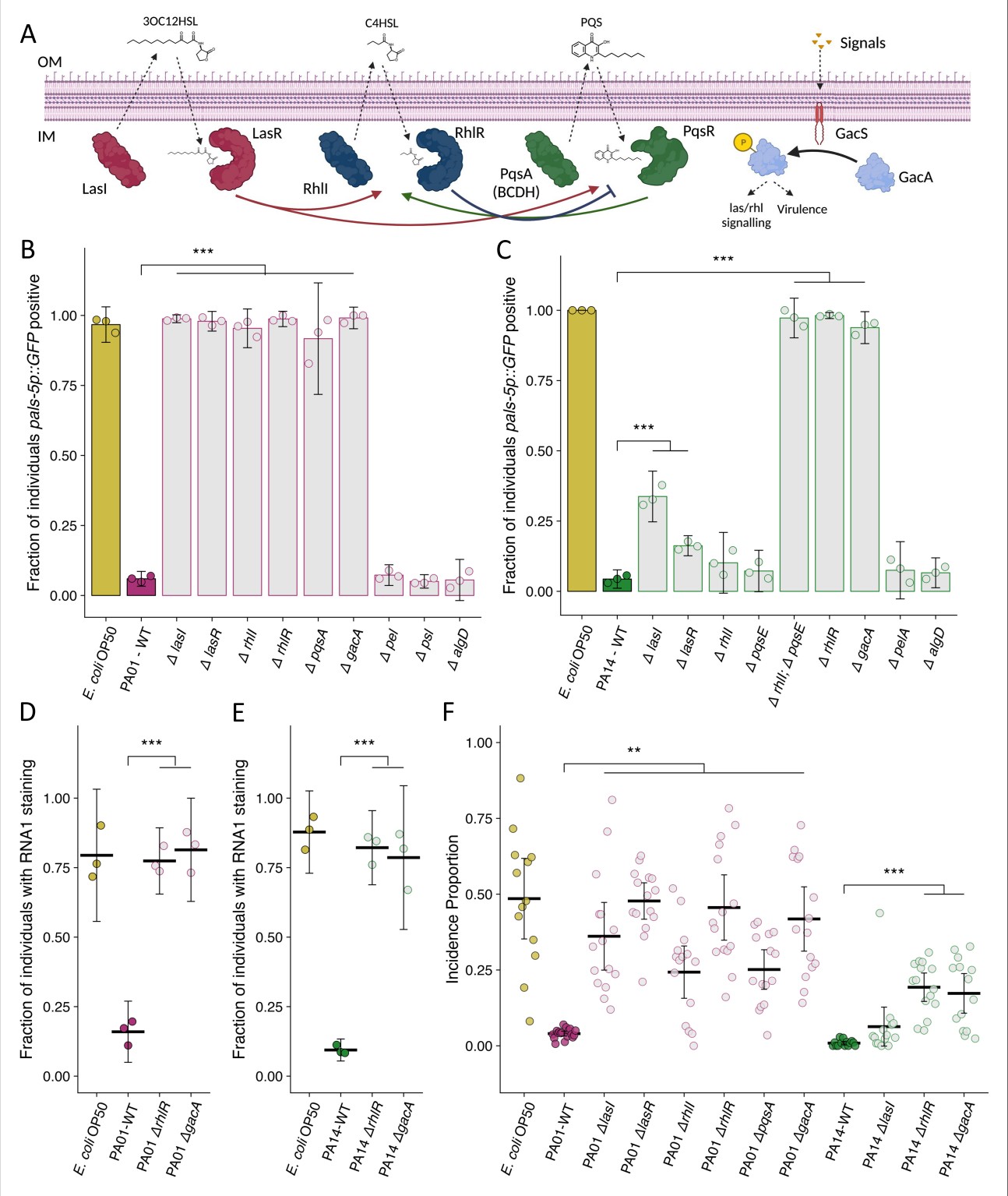

**Figure 4.** Mutation of *P. aeruginosa* quorum sensing regulators and the two-component response regulator *gacA* suppresses the attenuation of Orsay virus infection. (**A**) Diagram demonstrating the three quorum sensing systems in *P. aeruginosa*. Each system encodes the enzyme(s) (LasI, RhlI, PqsA(BCDH)) to produce an autoinducer (3OC12HSL – 3-oxo-C12-homoserine lactone, C4HSL – butanoyl homoserine lactone, PQS – *Pseudomonas* quinolone signal) that is recognized by its cognate receptor (LasR, RhlR, PqsR) that influences gene transcription and the activity of the other quorum sensing systems. An additional level of regulation stems from the two-component *gacS/gacA* system. External signals are recognized by the histidine kinase GacS, which phosphorylates the response regulator GacA that indirectly influences quorum sensing processes. IM is inner membrane, OM is

*Figure 4 continued on next page*

*Figure 4 continued*

outer membrane. Arrows represent crosstalk between the various system. Adapted from *Rutherford and Bassler, 2012* and *Song et al., 2023*. (**B–C**) The fraction of individuals that became *pals-5p::GFP* positive following exposure to exogenous Orsay virus in the presence of the indicated bacterium. Data are from a single representative experiment, bars represent mean, dots represent individual plates. (**B**) Wild-type *P. aeruginosa* PA01 compared to mutant *P. aeruginosa* PA01 strains using 10 a.u. of Orsay virus. Replicates can be found in *Figure 4—figure supplement 1* (n=4480 in total and n>76 for all dots). (**C**) Wild-type *P. aeruginosa* PA14 compared to mutant *P. aeruginosa* PA14 strains using 100 a.u. of Orsay virus. Replicates can be found in *Figure 4—figure supplement 2* (n=3320 in total and n>23 for all dots). (**D–E**) The fraction of individuals with staining following exposure to (**D**) 10 a.u. or (**E**) 100 a.u. Orsay virus as assessed by fluorescence in situ hybridization targeting the RNA1 segment of the Orsay virus genome. Data shown are from three experiments combined, each dot represents three pooled technical replicates from a single experiment. ((**D**) n=1632 in total and n>61 for all dots, (**E**) n=2000 in total and n>60 for all dots). (**F**) Incidence proportion of Orsay virus transmission in the presence of *P. aeruginosa* wild-type versus select *P. aeruginosa* mutants. Data are from three experiments, bars represent mean, dots represent individual plates (n=20,452 in total and n>37 for all dots). For all plots error bars represent 95% C.I. For B–E, p-values were determined using one-way ANOVA followed by Dunnett's test (NS, non-significant, *p<0.05, **p<0.01, ***p<0.001) (a.u., arbitrary units).

The online version of this article includes the following figure supplement(s) for figure 4:

**Figure supplement 1.** Replicate susceptibility assays shown in *Figure 4B*.

**Figure supplement 2.** Replicate susceptibility assays shown in *Figure 4C*.

**Figure supplement 3.** Supernatant from stationary phase cultures of different bacteria does not reliably attenuate or enhance Orsay virus infection in the presence of *E. coli* OP50.

similarly affected transmission from infected spreader animals. All *P. aeruginosa* PA01 quorum sensing mutants suppressed the attenuation of transmission by increasing the incidence proportion >6-fold compared to the wild-type *P. aeruginosa* PA01 (*Figure 4F*). *P. aeruginosa* PA14 *rhlR* and *gacA* mutants suppressed the attenuation of transmission by increasing the incidence proportion 19-fold and 17-fold respectively compared to wild-type *P. aeruginosa* PA14, but a *lasI* mutant had minimal effect consistent with the pattern we observed in the susceptibility assay using exogenous Orsay virus (*Figure 4F*). Likewise, the magnitude of suppression of the attenuation of Orsay virus transmission observed in the *P. aeruginosa rhlR* or *gacA* mutants was greater in the *P. aeruginosa* PA01 background compared to the *P. aeruginosa* PA14 background, potentially suggesting the existence of additional strain specific factors that act in *P. aeruginosa* PA14 to attenuate Orsay virus transmission.

Our findings suggest the possibility that quorum sensing molecules could act directly to attenuate infection. To explore this hypothesis, we prepared liquid *E. coli* OP50, *O. vermis* MYb71, *P. lurida* MYb11, *P. aeruginosa* PA01, and *P. aeruginosa* PA14 cultures and added culture supernatant to plates containing *E. coli* OP50 and Orsay virus. We did not observe any potent effect on host susceptibility to infection by Orsay virus from any supernatant (*Figure 4—figure supplement 3*), although it is difficult to rule out the possibility that the compounds may act at concentrations that are locally higher than the concentrations at which the compounds are present in our experiments.

## *P. aeruginosa* genes linked to virulence reduce Orsay virus transmission and infection

We designed a candidate-based screen using a non-redundant transposon insertion library to gain further insight into the genetic regulation of *P. aeruginosa* mediated reduction of Orsay virus transmission and infection (*Figure 5A*; *Liberati et al., 2006*). We identified candidate genes to include in our screen from three sources: *gacA*-regulated genes, *rhlR*-regulated genes (but *rhlI*- or *pqsE*-independent), and the set of genes required for full virulence in *C. elegans* previously identified by Feinbaum et al. (*Figure 5A*; *Brencic et al., 2009*; *Simanek et al., 2022*; *Feinbaum et al., 2012*). Of the 211 genes tested, 18 putative hits, including *lasI*, *rhlR*, and *gacA*, were identified with the corresponding *P. aeruginosa* PA14 mutants exhibiting suppression of the attenuation of Orsay virus infection by wild-type *P. aeruginosa* PA14 (*Figure 5A*, *Supplementary file 1c*). Using the 15 novel candidate genes, we performed an additional set of susceptibility assays which confirmed six of the hits (*Figure 5B*, *Figure 5—figure supplement 1*, *Supplementary file 1c*). The identified hits grouped into two clusters based on the strength of their suppression. Transposon insertion in the genes *ptsP*, *prpC*, and *kinB* led to marked suppression of the attenuation of Orsay virus leading to infection of 97%, 89%, and 82% of the population respectively compared to infection in only 4.2% of the

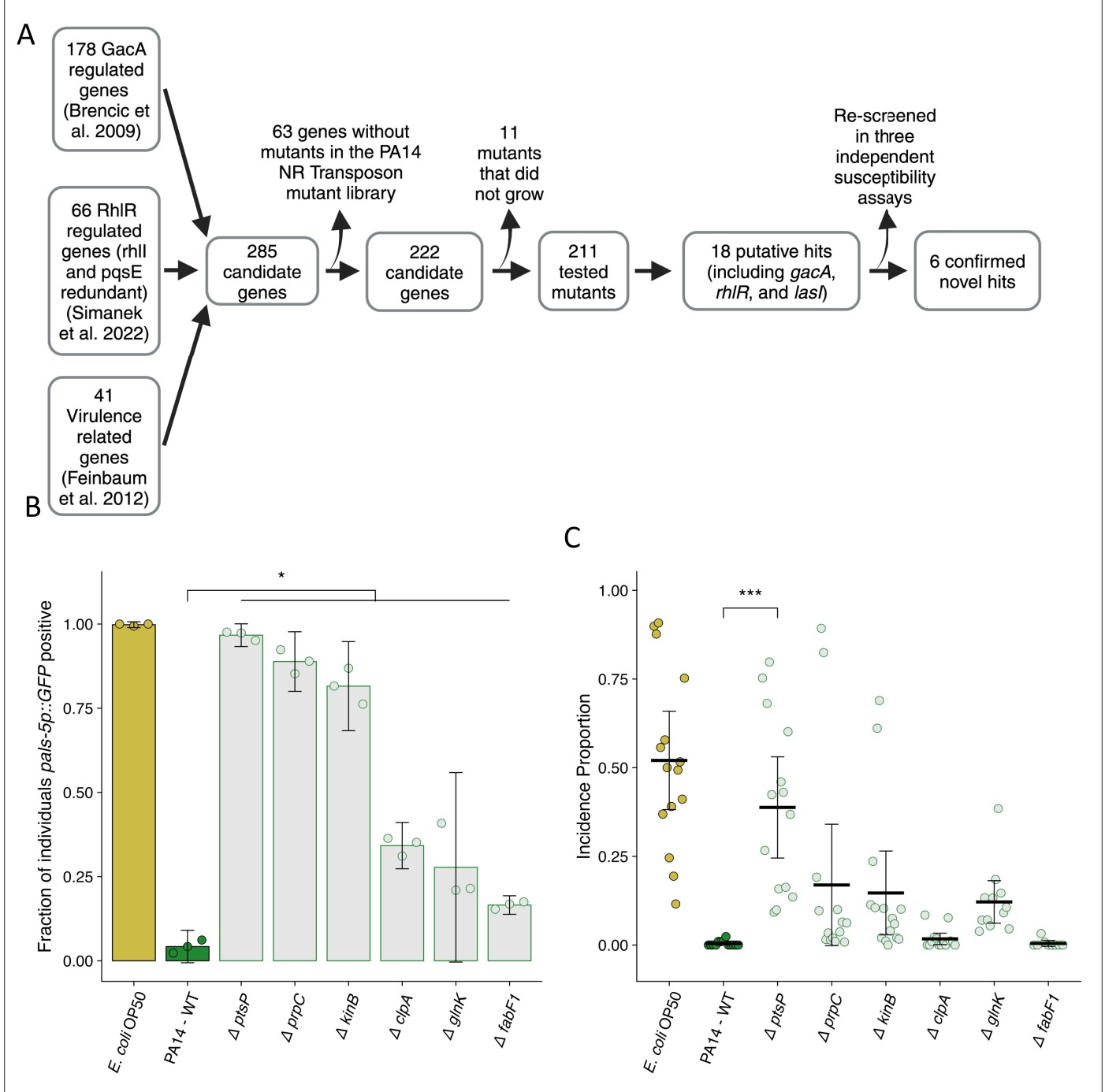

**Figure 5.** Mutation of *P. aeruginosa* PA14 genes related to virulence suppresses the attenuation of Orsay virus infection. (**A**) Diagram describing the design and results of the screen to identify suppressors of *P. aeruginosa* PA14 Orsay virus attenuation. (**B**) The fraction of individuals that became *pals-5p::GFP* positive following exposure to 100 a.u. of exogenous Orsay virus. Data are from a single representative experiment. Replicates can be found in *Figure 5—figure supplement 1* (n=3779 in total and n>55 for all dots). (**C**) Incidence proportion of Orsay virus transmission while individuals are present on *P. aeruginosa* PA14 wild-type compared to *P. aeruginosa* PA14 mutants. Data are from three experiments combined and dots represent individual plates (n=11,279 in total and n>35 for all dots). *E. coli* OP50 control shared with *Figure 6F*. For all plots the black bar is the mean and error bars are the 95% confidence interval (C.I.). p-Values determined using one-way ANOVA followed by Dunnett's test (NS, non-significant, *p<0.05, **p<0.01, ***p<0.001) (a.u., arbitrary units).

The online version of this article includes the following figure supplement(s) for figure 5:

**Figure supplement 1.** Replicate susceptibility assays shown in *Figure 5B*.

population in the presence of wild-type *P. aeruginosa* PA14 (*Figure 5B*, *Figure 5—figure supplement 1*). Transposon insertion in three additional genes, *clpA*, *glnK*, and *fabF1*, resulted in weaker, but robust suppression of the attenuation of Orsay virus leading to infection of 34%, 28%, and 17% of the population respectively (*Figure 5B*, *Figure 5—figure supplement 1*). We additionally tested whether transposon insertion into these genes suppressed attenuation of Orsay virus transmission. While we observed a trend toward mutations affecting susceptibility also affecting virus transmission, we observed a high degree of variation in the transmission assay, such that only mutation of *ptsP* led to statistically significant suppression of Orsay virus attenuation (*Figure 5C*).

We noted that all six of the hits originated from the set of genes required for full *P. aeruginosa* PA14 virulence in *C. elegans* (*Feinbaum et al., 2012*). These hits represented only 6 out of the 41 tested genes that are known to influence virulence, indicative of some degree of specificity in the consequences of these mutations beyond their general effect on *P. aeruginosa* PA14 virulence (*Feinbaum et al., 2012*).

## *P. lurida* MYb11 *gacA* is required for attenuation of Orsay virus transmission

We next sought to determine whether our findings from the interaction of *C. elegans* and Orsay virus in the presence of *P. aeruginosa* could inform us further regarding the mechanisms underlying the attenuation of Orsay virus transmission in the presence of *P. lurida* MYb11. In particular, we identified *gacA, ptsP, prpC, and kinB* orthologs using Orthovenn2 and generated a putative knockout allele of each gene by removing the coding potential for all but 10 amino acids from the N and C termini (*Figure 6A*; *Xu et al., 2019*). Following exposure to exogenous Orsay virus, knockout of *gacA* in *P. lurida* MYb11 led to infection of 77% of the population compared to infection of 43% of the population in the presence of wild-type *P. lurida* MYb11 from exogenous Orsay virus (*Figure 6B*, *Figure 6—figure supplement 1*). These observations were confirmed via FISH, as the *gacA* mutant supported infection of 62% of the population versus 37% infection in the presence of wild-type *P. lurida* MYb11 (*Figure 6C*). The *gacA* mutation also suppressed the attenuation of Orsay virus transmission, increasing the incidence proportion 2.9-fold compared to wild-type *P. lurida* MYb11 (*Figure 6D*). These data suggest that *gacA* has a conserved role between *P. aeruginosa* and *P. lurida* MYb11 in the attenuation of Orsay virus transmission and infection of *C. elegans*. On the other hand, knockout of the *P. lurida* MYb11 orthologs of *ptsP*, *prpC*, or *kinB* failed to suppress the attenuation of Orsay virus infection or transmission by *P. lurida* MYb11 (*Figure 6E,F*, *Figure 6—figure supplement 1C, D*; *Feinbaum et al., 2012*; *Sun et al., 2023*). One explanation of these results is that these genes play different roles within *P. lurida* MYb11 and *P. aeruginosa* and therefore their mutation did not have the same consequences for Orsay virus attenuation. Alternatively, these results might suggest that *ptsP*, *prpC*, and *kinB* act to attenuate Orsay virus infection via some specific effect on *P. aeruginosa* PA14 virulence but do not influence the interaction of *P. lurida* MYb11 with *C. elegans*.

## *O. vermis* MYb71 closely interacts with the *C. elegans* intestinal brush border

*P. lurida* MYb11 and *O. vermis* MYb71 can both be isolated from *C. elegans* natural environment (*Dirksen et al., 2016*; *Dirksen, 2020*). We were therefore curious how exposure to both bacteria simultaneously would impact Orsay virus infection rates. When cultures of *O. vermis* MYb71 and *P. lurida* MYb11 were concentrated to 25 mg/mL and then mixed, we observed that a 50% mixture of each bacterium (vol:vol) attenuated infection to the same degree as pure *P. lurida* MYb11 (*Figure 7A*, *Figure 7—figure supplement 1A and B*). Moreover, we also noted that mixing *O. vermis* MYb71 with *E. coli* OP50 also lead to infection rates similar to pure *E. coli* OP50 (*Figure 7A*, *Figure 7—figure supplement 1A and B*). Previous studies have identified that *Ochrobactrum* is found in large numbers in the *C. elegans* intestine following exposure (*Dirksen et al., 2016*; *Dirksen, 2020*). We investigated how the levels of *Ochrobactrum* in the intestine changed following exposure to pure or mixed lawns of bacteria. When alone, GFP expressing *Ochrobactrum* BH3 was readily observed within the intestine at 4 hr or 24 hr of exposure. Interestingly, exposure to a mixed lawn of *Ochrobactrum* BH3 and *P. lurida* MYb11 reduced intestinal accumulation of *Ochrobactrum* BH3 compared to the pure *Ochrobactrum* BH3 treatment (*Figure 7B and C*, and *Figure 7—figure supplement 1C–F*). On the other hand, exposure to a mixed lawn of *Ochrobactrum* BH3 and *E. coli* OP50 reduced accumulation

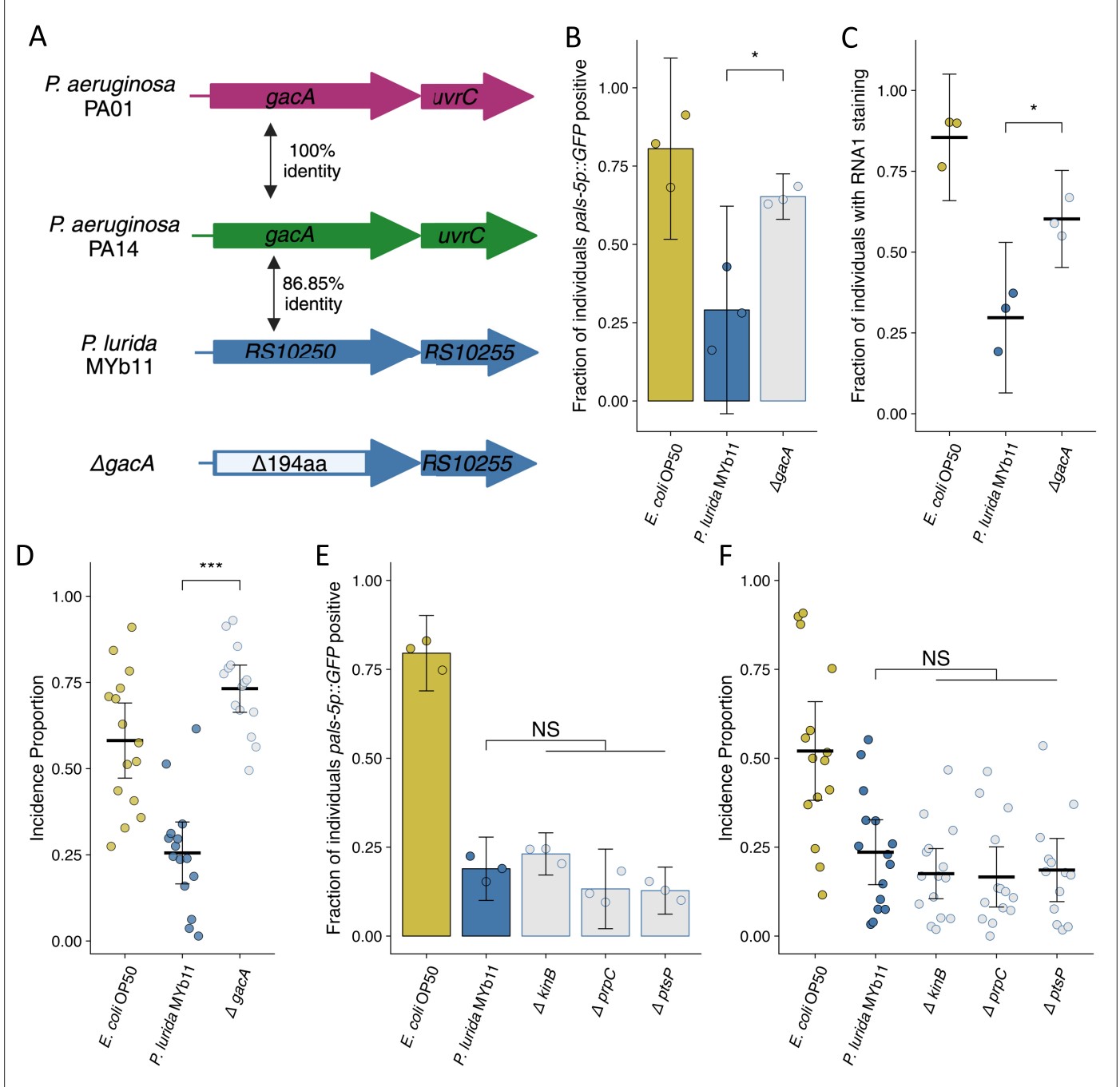

**Figure 6.** Mutation of *P. lurida* MYb11 *gacA* suppresses the attenuation of Orsay virus infection. (**A**) Diagram depicting *P. aeruginosa* PA01, *P. aeruginosa* PA14, and *P. lurida* MYb11 *gacA* as well as the *P. lurida* MYb11 *gacA* deletion mutant. Percent identity of the encoded protein was assessed using Clustal Omega. The *gacA* deletion removed 194 amino acids from the protein leaving 10 amino acids from both the N and C termini. (**B, E**) The fraction of individuals that became *pals-5p::GFP* positive following exposure to 10 a.u. exogenous Orsay virus in the presence of wild-type *P. lurida* MYb11 compared to the indicated mutants. Data are from a single representative experiment. Replicates can be found in *Figure 6—figure supplement 1*. ((**B**) n=933 in total and n>46 for all dots, (**E**) n=1725 in total and n>78 for all dots). Bars represent mean and dots represent individual plates. (**C**) The fraction of individuals with staining following exposure to 10 a.u. Orsay virus as assessed by fluorescence in situ hybridization targeting the RNA1 segment of the Orsay virus genome. Data shown are from three experiments combined, each dot represents three pooled technical replicates from (**B**) and *Figure 6—figure supplement 1* (n=1272 in total and n>61 for all dots). (**D, F**) Incidence proportion of Orsay virus transmission in the presence of wild-type *P. lurida* MYb11 versus the indicated mutants. Data are from three experiments combined, bars represent mean, dots represent individual plates ((**D**) n=4534 in total and n>42 for all dots, (**F**) n=8185 in total and n>58 for all dots). *E. coli* OP50 control of Figure 6F shared with *Figure 5C*. For

*Figure 6 continued on next page*

*Figure 6 continued*

all plots error bars represent 95% confidence interval (C.I.). (**B–D**) p-Values were determined using Student's t-test. (**E–F**) p-Values determined using one-way ANOVA followed by Dunnett's test (NS, non-significant, *p<0.05, **p<0.01, ***p<0.001) (a.u., arbitrary units).

The online version of this article includes the following figure supplement(s) for figure 6:

**Figure supplement 1.** Replicate susceptibility assays shown in *Figure 6B and E*.

of *Ochrobactrum* BH3 at 4 hr compared to a pure *Ochrobactrum* BH3 treatment, but this difference was minimal by 24 hr (*Figure 7B and C*, and *Figure 7—figure supplement 1C–F*).

These data suggested that the presence of *Ochrobactrum* in the intestine may be important for its ability to promote Orsay virus infection. We therefore fixed *C. elegans* exposed to *O. vermis* MYb71 or *E. coli* OP50 for examination via electron microscopy and imaged cross sections of the intestine. As expected from our fluorescence experiments, *O. vermis* MYb71 was readily observed in the intestinal lumen at 4 hr or 24 hr of exposure (*Figure 7D and E*). The glycocalyx is the mucus-like layer covering *C. elegans* intestinal cells and that recent work has suggested may play a role in defense against Orsay virus (*McGhee, 2007*; *Zhou et al., 2024*). The glycocalyx of animals exposed to *O. vermis* MYb71 was observed with regions of variable thickness and potential instances of *O. vermis* MYb71 deforming the microvilli brush border as opposed to animals exposed to *E. coli* OP50 where the glycocalyx appeared uniform and undisturbed (*Figure 7D and E*). A recent report observed that *S. marcescens* promoted infection of the mosquito *A. aegypti* by Dengue, Zika, and Sindbis viruses by secreting a protein, enhancin, that degrades the mucus layer covering epithelial cells (*Wu et al., 2019*). These observations lead us to speculate that *O. vermis* MYb71 mediated disruption of the brush border and glycocalyx promotes Orsay virus infection although this hypothesis has yet to be tested.

## Discussion

In our study, we quantitatively define the effect of bacteria on the transmission of Orsay virus in the *C. elegans* host. While the presence of *O. vermis* MYb71 enhanced the transmission of Orsay virus, the presence of *P. lurida* MYb11 or *P. aeruginosa* strains PA01 and PA14 attenuated Orsay virus transmission. The enhancement of Orsay virus transmission by *O. vermis* MYb71 and reduction of transmission by *P. lurida* MYb11 and *P. aeruginosa* PA01 and PA14 was mirrored in assays assessing infection rates by exogenous virus. Our results using exogenous virus demonstrate that host susceptibility to Orsay virus infection may vary by over three orders of magnitude in the presence of *O. vermis* MYb71 versus *P. lurida* MYb11 which are found in the natural environment in association with *C. elegans* (*Dirksen et al., 2016*; *Dirksen, 2020*). Moreover, we observe that pathogenic *P. aeruginosa*, which may be found in natural environments, can further attenuate Orsay virus transmission (*Crone et al., 2020*). Our work is also consistent with a recent study by González and Félix that also reported that monoaxenic cultures of other bacteria from the environment of *C. elegans* impact host susceptibility (*González and Félix, 2024*).

Our observations that two *Ochrobactrum* species promoted transmission of Orsay virus are intriguing given that other *Ochrobactrum* species have also been linked to viral infections. *Ochrobactrum intermedium* promoted poliovirus infection in mice and enhanced poliovirus stability in vitro (*Kuss et al., 2011*). *Ochrobactrum anthropi*, an opportunistic human pathogen, was identified using a random forest analysis as one of the most predictive features differentiating the upper respiratory tract of human patients recovering from influenza infection versus healthy controls (*Kaul et al., 2020*). Our observations in the *C. elegans* host raise the speculative possibility that *Ochrobactrum* colonization may have roles in evolutionarily diverse hosts in modulating viral infection. Further, our electron micrographs point to an intriguing hypothesis that *Ochrobactrum* disrupts the brush border region of epithelial cells thereby promoting viral infection. Such a mechanism would be consistent with recent work identifying that *S. marcescens* produces the protein enhancin, which promotes viral infection by degrading the mucus layer covering epithelial cells in the mosquito *A. aegypti* (*Wu et al., 2019*). Additional work will need to be performed to explicitly test whether such a mechanism is at play in *Ochrobactrum* mediated enhancement of Orsay virus infection and transmission.

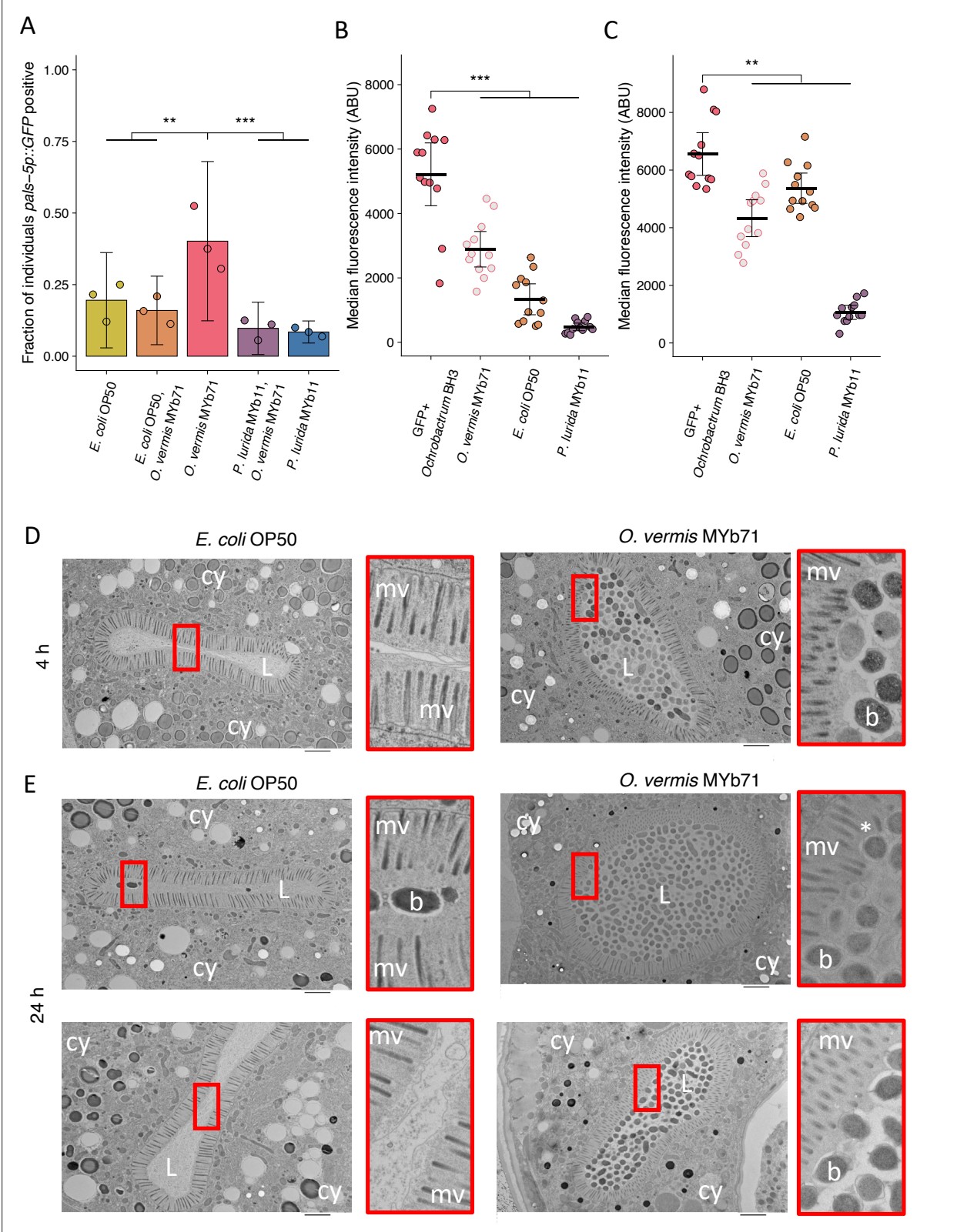

**Figure 7.** *O. vermis* MYb71 closely interacts with the brush border of *C. elegans* intestinal cells. (**A**) The fraction of individuals that became *pals-5p::GFP* positive following exposure to 1 a.u. exogenous Orsay virus in the presence of pure lawns of the indicated bacteria or mixed lawns consisting of 50% of the indicated bacteria mixed with 50% *O. vermis* MYb71 (vol:vol, Materials and methods). Data are from a single representative experiment. Replicates can be found in *Figure 7—figure supplement 1*. Bars represent mean and dots represent individual plates (n=908 in total and n>36 for all

*Figure 7 continued on next page*

*Figure 7 continued*

dots). (**B–C**) The median fluorescence observed in the intestinal lumen of individuals following exposure for (**B**) 4 hr or (**C**) 24 hr to pure GFP expressing *Ochrobactrum* BH3 lawns or mixed lawns consisting of 50% of the indicated bacteria mixed with 50% GFP expressing *Ochrobactrum* BH3 (vol:vol, Materials and methods). Data are from a single representative experiment. Replicates can be found in *Figure 7—figure supplement 1*. Bars represent mean, dots represent individual animals. (**D–E**) Electron micrographs of animals exposed to the indicated bacterium for either (**D**) 4 hr or (**E**) 24 hr. Scale bar is 2 μm. cy = cytoplasm, L=lumen, mv = microvilli, b=bacterium, and * denotes a bent microvilli. For all plots error bars represent 95% confidence interval (C.I.). p-Values determined using one-way ANOVA followed by Dunnett's test (NS, non-significant, *p<0.05, **p<0.01, ***p<0.001) (a.u., arbitrary units).

The online version of this article includes the following figure supplement(s) for figure 7:

**Figure supplement 1.** Replicate assays shown in *Figure 7A–C*.

---

*P. lurida* MYb11 and *P. aeruginosa* PA01 and PA14 shared the ability to attenuate Orsay virus infection and transmission. *P. lurida* MYb11 promotes developmental rate and fitness at a cost to overall lifespan (*Dirksen, 2020*; *Kissoyan et al., 2022*). In contrast, *P. aeruginosa* rapidly kills *C. elegans* and is detrimental to host fitness (*Tan et al., 1999a*; *Rahme et al., 1995*). Under our experimental conditions exposure to all three shortened lifespan compared to exposure to *E. coli* OP50. Given that *O. vermis* MYb71 shortened lifespan to the same extent as *P. aeruginosa* PA14 but had the opposite effect on host susceptibility to Orsay virus, we do not believe that general virulence by any bacterium is responsible for the observed effects on host susceptibility. Rather, unique virulence processes specific to each bacterium may contribute to the unique effects of each bacterium on host susceptibility to Orsay virus.

We identified that *gacA* regulates the attenuation of Orsay virus by *P. aeruginosa* PA01 and PA14 and *P. lurida* MYb11. In *Pseudomonas*, *gacA/gacS* regulate processes related to quorum sensing and virulence (*Lapouge et al., 2008*). Quorum signaling influences additional aspects of *P. aeruginosa* physiology, including swarming behaviors, biofilm formation, secondary metabolism rates, and overall transcription patterns (*Lee and Zhang, 2015*). To gain additional insights into the *P. aeruginosa* PA14 effects on Orsay virus transmission we conducted a candidate-based screen of genes influenced by quorum signaling and genes that regulate virulence toward *C. elegans* (*Brencic et al., 2009*; *Simanek et al., 2022*; *Feinbaum et al., 2012*). We identified six additional genes, *ptsP*, *prpC*, *kinB*, *clpA*, *glnK*, and *fabF1*, that when mutated suppressed *P. aeruginosa* PA14 attenuation of Orsay virus infection. Each of these genes, in addition to *gacA*, *rhlR*, and *lasI*, has been shown to be required for full *P. aeruginosa* PA14 virulence in *C. elegans* (*Feinbaum et al., 2012*). However, we observed that only these 6 out of the total 41 virulence-related genes identified in Feinbaum et al. influenced Orsay virus infection, arguing further against a role for general virulence in the attenuation of Orsay virus transmission and infection caused by *P. aeruginosa* PA14 and *P. aeruginosa* PA01 (*Feinbaum et al., 2012*). Additionally, while mutation of the *ptsP*, *prpC*, and *kinB* orthologs of *P. lurida* MYb11 failed to suppress the attenuation of Orsay virus, it is unclear whether these genes influence the effect of *P. lurida* MYb11 on host lifespan or even whether *P. lurida* MYb11 pathogenicity mediated by other factors leads to attenuation of Orsay virus infection.

*C. elegans* is unlikely to associate with a single bacterium in its natural environment. However, *C. elegans* shows clear behavioral preferences for grazing on certain bacteria from its environment and may eat monoxenic lawns in the wild (*Shtonda and Avery, 2006*). Our data demonstrate that individual bacteria can have a profound impact on Orsay virus transmission rates in a species-specific manner. Additionally, the tractability of the *C. elegans*-Orsay virus experimental system allowed us to identify molecular determinants of viral transmission and will be useful for identifying additional biotic factors that influence viral transmission. Our work builds on the expanding body of knowledge showing that the microbiota can influence the interactions between viruses and their animal hosts.

## Materials and methods
### *C. elegans* strains and growth conditions

*C. elegans* were maintained on NGM agar plates (17 g agar, 2.5 g peptone, 3 g NaCl per 1 L water) containing *E. coli* OP50 (*Brenner, 1974*). Strains bearing the *glp-4(bn2)* temperature-sensitive mutation were maintained at 16°C, while all other were maintained at 20°C. All assays were performed on SKA assay plates at 20°C (17 g agar, 3.5 g peptone, 3 g NaCl per 1 L water) (*Tan et al., 1999a*). A full

list of *C. elegans* strains used in this study is contained within the Supplemental Information (*Supplementary file 1a*). For all assays relying upon *pals-5p::GFP*, induction was manually assessed using a Nikon SMZ18 stereofluorescent microscope.

## Bacterial strains and growth conditions

All bacteria were grown in Luria broth (10 g tryptone, 5 g yeast, 10 g NaCl per 1 L water). *E. coli* OP50 and *P. aeruginosa* strains were grown at 37°C with shaking, while all other strains were grown at 27°C with shaking. A full list of bacterial strains used in this study is contained within the Supplemental Information (*Supplementary file 1b*).

## Orsay virus isolation and batch testing

Orsay virus was isolated from infected WUM31(*rde-1(ne219);jyIs9[pals-5p::gfp;myo-2p::mCherry]*) individuals. Plates with gravid WUM31 were bleached to obtain eggs. Eggs were hatched overnight in M9 solution rotating at 20°C and the L1 larvae were arrested to synchronize the population. L1 larvae were combined with 100 µL of 6× concentrated *E. coli* OP50 liquid culture and 50 µL of Orsay virus filtrate, plated on SKA plates, and once dried, placed at 20°C for 48 hr. Four GFP-positive individuals were then transferred to a new 6 cm NGM plate with *E. coli* OP50 and maintained until just starved. Twenty such 6 cm plates were washed with 10 mL of M9 and the resulting suspension was Dounce homogenized. Alternatively, two 3.5 cm SKA plates containing infected WUM31 animals were allowed to starve and then equally chunked onto eight 10 cm NGM plates with *E. coli* OP50. Once these plates were just starved, the plates were washed with 10 mL of M9 and homogenized as above. The homogenized suspension was then centrifuged at 13,200×*g* for 5 min. The supernatant was passed through a 0.22 µm filter and aliquoted. Each batch of virus was tested for potency and an $ID_{50}$ calculated for ZD2611 populations in the presence of *E. coli* OP50 (*Figure 1—figure supplement 2*). On average, 3.6 µL of Orsay virus stock prepared as indicated in the Materials and methods was required to infect 50% of a population of ZD2611 animals in the presence of *E. coli* OP50 after 24 hr (*Figure 1—figure supplement 2*). We observed variability in the measured $ID_{50}$ even when using the same batch of Orsay virus (e.g. *Figure 1D* vs *Figure 1—figure supplement 3* vs. *Figure 1—figure supplement 3*). We therefore chose to control all our experiments internally rather than attempt to normalize between Orsay virus batches and we report doses used in a.u., however 1 a.u. corresponds to 1 µL of Orsay virus filtrate.

## Preparation of uninfected individuals for transmission and susceptibility assays

Prior to all assays, plates with fecund ZD2611(*glp-4(bn-2);jyIs8[pals-5p::gfp;myo-2p::mCherry]*) were bleached to obtain eggs. Eggs were hatched overnight in M9 solution rotating at 20°C and the L1 larvae were arrested to synchronize the population. ZD2611 L1s were dropped onto plates containing *E. coli* OP50 and placed at 20°C for 64–72 hr. At this temperature reproduction is delayed, but not eliminated, and no L1s are observed during transmission or susceptibility assays. The resulting young adults were washed off the plates in M9 and centrifuged at 1000×*g* for 1 min. After removing the supernatant, the young adults were once again washed in M9, then centrifuged at 1000×*g* for 1 min. Young adults were directly dropped onto prepared plates described below.

## Preparation of bacteria for transmission and susceptibility assays

Unless otherwise indicated, bacteria were cultured overnight and added to cover the assay plates. Plates containing *E. coli* OP50 or *P. aeruginosa* strains were placed at 37°C for 24 hr. Plates containing *O. vermis* MYb71 or *P. lurida* MYb11 strains were placed at 25°C for 24 hr. All plates were then moved to room temperature (RT) for an additional 24 hr before the addition of virus.

For the natural isolate transmission screen and early transmission assay cultures were grown to late stationary phase. Cultures were then spun down for 6 min at 4000×*g* and reconstituted to 25 mg/mL in their own supernatant. 150 µL of this suspension was combined with 50 µL of a mix of M9 for transmission assays or 50 µL of a combination of Orsay virus filtrate and M9 for the early transmission assay. This mixture was combined with young adult ZD2611 individuals and directly added to cover assay plates before drying.

## Transmission assays

To obtain infected spreader animals, fecund ZD2610 (*glp-4(bn-20); rde-1(ne219); jyIs8[pals-5p::gfp;myo-2p::mCherry]*) were bleached to obtain eggs. Eggs were hatched overnight in M9 solution rotating at 20°C and the L1 larvae were arrested to synchronize the population. Arrested ZD2610 L1s were dropped onto NGM plates containing *E. coli* OP50 and placed at 20°C for 48 hr. Animals were then washed off the plates in M9 and centrifuged at 1000×*g* for 1 min. After removing the supernatant, the young adults were washed with M9 again then centrifuged at 1000×*g* for 1 min. Supernatant was once again removed, and individuals were immediately mixed with 100 µL of 6× overnight *E. coli* OP50 culture and 50 µL of Orsay virus for a minimum of 18–20 hr. *pals-5p::GFP*-positive ZD2610 young adults were identified and picked onto transfer plates containing the appropriate bacteria prepared as indicated above for 4–6 hr. After this period, five spreaders were transferred to assay plates containing the same bacteria and approximately 100 uninfected ZD2611 individuals to start the assay. Transmission assays were scored 24 hr after adding spreader individuals.

For the natural isolate transmission screen, arrested ZD2610 L1s were combined with 100 µL of 6× concentrated *E. coli* OP50 and 50 µL of Orsay virus filtrate and once dried, placed at 20°C for 64–72 hr. On the day of the assay, *pals-5p::GFP*-positive ZD2610 young adults were identified and picked onto transfer plates containing the appropriate bacteria prepared as indicated above for 4–6 hr. After this period, five spreaders were transferred to assay plates containing the same bacteria and approximately 100 uninfected ZD2611 individuals to start the assay.

## Incidence proportion

Incidence proportion was calculated by dividing the number of newly infected animals by the total non-spreader population per plate. Individual plates for which the total number of *pals-5p::GFP*-positive individuals after 24 hr was less than the initial number of spreaders placed on the plate were excluded to eliminate potential differences in incidence proportion caused by unequal numbers of infected spreader animals. For *Figure 1C* a single plate was censored from *P. lurida* MYb11, for *Figure 2A* six plates were censored from *P. aeruginosa* PA14, and for *Figure 4E* 12 plates were censored: seven from *P. aeruginosa* PA14, three from *P. aeruginosa* PA01, one from *P. aeruginosa* PA14Δ*gacA*, and one from *P. aeruginosa* PA14Δ*lasI*.

## Susceptibility assays

Unless otherwise indicated, Orsay virus filtrate at the indicated doses was diluted in filtered M9 solution. 200 µL total was applied to each assay plate and swirled to cover the entire bacterial lawn. Plates were then dried before the addition of approximately 100 uninfected ZD2611 individuals prepared as indicated above. Assays were scored 24 hr later using a Nikon SMZ18 stereo fluorescent microscope and the fraction of *pals-5p::GFP*-positive individuals was calculated.

## Early transmission assay

For early transmission assay, young ZD2611 adults were exposed to 0 a.u. or 5 a.u. exogenous Orsay virus in the presence of *O. vermis* MYb71. 8 hr post infection five infected individuals from the 5 a.u. plate were transferred to the 0 a.u. plate to assess whether transmission from these individuals would occur. Both plates were scored 16 hr after the transfer.

## Supernatant transfer assay

Assay plates were prepared with *E. coli* OP50 for susceptibility assays as indicated above. Bacteria were inoculated into 6 mL of LB and allowed to grow for 48 hr. Cultures were then spun at 4000×*g* for 6 min and the spent media supernatant was passed through a 0.22 µm filter. 200 µL of the filtered supernatant was added to the *E. coli* OP50 assay plates just prior to the addition of Orsay virus and ZD2611 young adults.

## Fluorescence in situ hybridization

Previously published probes were obtained to target the RNA1 segment of Orsay virus (*Franz et al., 2014*). Animals were infected according to the methods described above for susceptibility assays. Animals were processed according to the Stellaris RNA FISH Protocol for *C. elegans* (LGC Biosearch Technologies) with minor modifications. Briefly, young adults were washed off the plate and rinsed

twice in M9. Animals were then fixed for 30 min rotating in a microcentrifuge tube at 20°C. After washing twice with 1 mL of phosphate buffered saline animals were permeabilized in 70% of ethanol and stored at 4°C for 1–7 days. Animals were washed with Wash Buffer A before addition of 100 µL of hybridization buffer containing 3 µL of RNA1 probe mix. Probe was hybridized overnight at 46°C. Animals were then washed with Wash Buffer A alone once, and then again with Wash Buffer A containing 5 ng/mL DAPI. Lastly 100 µL of Wash Buffer B was added before mounting animals on slides with 25 µL of Vectashield Mounting Medium. The fraction of individuals with RNA1 staining was then quantified using a Nikon SMZ18 stereofluorescent microscope or a Zeiss AxioImager Z1 compound fluorescent microscope.

## Viral stability

Assay plates with each bacterium were prepared as described above. 50 a.u. Orsay virus was added to each plate as in a susceptibility assay, however no *C. elegans* were added. Plates were washed with 1 mL of ddH$_2$O after 30 min or 24 hr. A spike-in control was also included where 50 a.u. of Orsay virus was directly added to 1 ml H$_2$O and processed the same as all other samples. The resulting mixture was spun at 15,000 rpm (max speed) at 4°C for 10 min. 1 µL of the supernatant was used to make cDNA (Promega GoScript reverse transcriptase using random primers (A2801)). Orsay virus RNA1 level was quantified by qPCR using 1 µL of 1/5 diluted cDNA (GoTaq Promega A6001) and run on QuantStudio 3 Real Time PCR system (RNA1 qPCR primers GW194 and GW195) (*Félix et al., 2011*).

## Viral shedding

Infected ZD2610 animals were prepared as indicated above with infection occurring at the L4 stage. 24 hr later as young adults, 20 *pals-5p::GFP*-positive animals were transferred to assay plates prepared with bacteria as indicated above for a transmission assay. These animals were allowed to shed virus onto the plate for 24 hr at 20°C. Spreaders were then removed and the assay plates were washed with 1 mL of M9. The resulting mixture was spun at 15,000 rpm (max speed) at 4°C for 10 min. 1 µL of the supernatant was used to make cDNA (Promega GoScript reverse transcriptase using random primers (A2801)). Orsay Virus RNA1 level was quantified by qPCR using 1 µL of 1/5 diluted cDNA (GoTaq Promega A6001) and run on QuantStudio 3 Real Time PCR system (RNA1 qPCR primers GW194 and GW195) (*Félix et al., 2011*).

## RNA extraction

Animals were washed five times in M9 and collected in TRIzol reagent (Invitrogen) and stored at –80°C before extraction. RNA extraction was performed using Direct-zol RNA microprep kits (Zymo Research) following the manufacturer's instructions.

## qPCR

cDNA was made using 1000 ng of RNA as template (Promega GoScript Reverse Transcriptase using Random Primers). cDNA was diluted at 1/80 and qPCR were performed using 1 µL of diluted cDNA (GoTaq Promega) and run on QuantStudio 3 Real Time PCR system. RNA1 levels were then quantified via qPCR using previously published primers (*Félix et al., 2011*). Orsay virus RNA levels were normalized to the internal control host gene snb-1 (primers GCTCAGGTTGATGAAGTCGTC and GGTG GCCGCAGATTTCTC).

## Plasmid-based replication experiments

Adult animals carrying a transgene containing the wild-type RNA1 or RNA1D601A Orsay virus genome segment under the control of a heat-inducible promoter were placed on the indicated bacteria for 4 hr before heat-shock at 33°C for 2 hr (*Jiang et al., 2017*). Animals then recovered at 20°C for 20 hr before harvesting for RNA extraction and qPCR as detailed above. RNA1 levels were normalized to the values obtained from animals bearing the wild-type RNA1 and exposed to *E. coli* OP50.

## Orsay virus replication in the presence of *P. lurida* MYb11 and *P. aeruginosa*

Adult ERT54 (*jyIs8[pals-5p::gfp;myo-2p::mCherry]*) or WUM31 (*rde-1(ne219); jyIs8[pals-5p::gfp;myo-2p::mCherry]*) were exposed to exogenous Orsay virus in the presence of *P. lurida* MYb11, *P. aeruginosa*

PA01, or *P. aeruginosa* PA14 prepared as in a susceptibility assay. After 2 hr or 24 hr, animals were harvested for RNA extraction and qPCR as detailed above. RNA1 levels were quantified via qPCR. Within each genotype, the data for each experiment were normalized to the 2 hr timepoint.

### *P. lurida* MYb11 mutant construction

A protocol developed for allelic exchange in *P. aeruginosa* was modified for use in *P. lurida* MYb11 (*Hmelo et al., 2015*). Briefly, homology arms flanking the region to be deleted were obtained using polymerase chain reaction (PCR) and cloned into the pExG2-KanR suicide vector using Hi-Fi Assembly (New England Biolabs) (*Rietsch et al., 2005*). DH5α *E. coli* were transformed using a standard heat-shock protocol. Successful transformants were selected for LB+Kanamycin (50 μg/mL) plates and colony PCR was performed to check for proper insert size in the transformants. *E. coli* bearing the desired plasmid were grown overnight in LB+Kanamycin (50 μg/mL). Plasmids were then obtained using a QIAGEN MiniPrep Kit (QIAGEN). Plasmids were assessed for the desired sequence by Sanger sequencing and transformed into *P. lurida* MYb11 using the following electroporation procedure. *P. lurida* MYb11 was grown overnight then placed on ice for 30 min. The culture was spun down at 4°C and washed twice with ice-cold water. After reconstitution in 100 μL of ice-cold water the suspension was transferred to a pre-chilled cuvette and transformed at 630 kV using an Eporator (Eppendorf). 900 μL of LB was added, and the suspension was transferred to a microcentrifuge tube and incubated at 27°C for 2 hr with shaking. The suspension was then plated on LB+Kanamycin (50 μg/mL) plates and grown for 48 hr at 25°C. Colonies were picked and grown overnight in LB. Cultures were streaked onto sucrose plates (15 g agar, 10 g tryptone, 5 g yeast, 60 g sucrose per 1 L water) to perform sucrose-based counter selection (*Hmelo et al., 2015*). Colonies that survived were genotyped for the expected deletion.

### Lifespan analysis

Assay plates, bacteria, and ZD2611 animals were prepared as indicated above, however plates were prepared with the addition of 50 μg of 5-fluorodeoxyuridine to eliminate the need to transfer animals during the assay by preventing reproduction. 30 young adult ZD2611 animals were added to each assay plate and assessed for survival every 24 hr. Three replicate plates were prepared per experiment and three experiments were performed in total.

### Pumping rate

Assay plates, bacteria, and ZD2611 animals were prepared as indicated above. 20 young adult ZD2611 animals were transferred to each assay plate. At 6 hr and 24 hr, animals were recorded with slow-motion video capture (1/4×) while feeding. Cycles of pharyngeal pumping were counted and used to calculate the number of pharyngeal pumps per minute.

### Mixing susceptibility assays and colonization assays

Bacteria were grown in LB for 48 hr before being spun down at 4000×*g* for 6 min. The resulting cell pellets were weighed and reconstituted to 25 mg/mL in their own culture supernatant. Concentrated bacteria were then combined with filtered M9 for colonization experiments or filtered M9 plus 1 μL of Orsay filtrate for mixing susceptibility assays. For mixing susceptibility assays young adult ZD2611 individuals prepared as above were added to the mixture before addition to plates such that the bacterial lawn covered the entire plate. For colonization assays N2 animals were prepared in the same manner to adulthood. Mixing susceptibility assays were scored as a susceptibility assay as indicated above. Colonization assays were scored at 4 hr or 24 hr. Briefly, animals were picked off of the plate and anesthetized in sodium azide before mounting on a microscope slide. Fluorescent images of the region of the intestinal lumen behind the pharynx were imaged using a Zeiss AxioImager Z.1 compound microscope. Images were then imported into FIJI for analysis (*Schneider et al., 2012*; *Schindelin et al., 2012*). An irregular polygon tracing the region behind the pharynx was drawn and the median fluorescence intensity measured.

### Electron microscopy

Animals were prepared for electron microscopy as indicated above for susceptibility assays. Animals were washed off their plates in M9 and allowed to pellet by gravity. The supernatant was removed and

animals were anesthetized with sodium azide. Animals were then pipetted into a type A 6 mm Cu/Au carrier (Leica) and frozen in a high-pressure freezer (EM ICE, Leica). This was followed by the following freeze substitution protocol: (EM AFS2, Leica).

## Freeze substitution protocol
Cocktail 1: 2% osmium tetroxide, 0.1% uranyl acetate in anhydrous acetone.

## Program

| Cocktail | Temperature (start) | Temperature (end) | Time |
|---|---|---|---|
| Cocktail 1 | –90°C | –90°C | 60 hr |
| Cocktail 1 | –90°C | –20°C | 11 hr |
| Cocktail 1 | –20°C | –20°C | 10 hr |
| Cocktail 1 | –20°C | 0°C | 12 hr |
| Acetone wash (4×) | 0°C | 22°C | 2 hr |

After the washes in acetone, samples were incubated in propylene oxide for 30 min, then slowly infiltrated in TAAB Epon (TAAB Laboratories Equipment Ltd, https://taab.co.uk) as follows:

3:1 Propylene Oxide:TAAB Epon – 3 hr at RT
1:1 Propylene Oxide:TAAB Epon – ON at 4°C
1:3 Propylene Oxide:TAAB Epon – 4 hr at RT
100% TAAB Epon – 2 hr at RT

Samples were embedded in fresh TAAB Epon and polymerized at 60°C for 48 hr.

Ultrathin sections (about 80 nm) were cut on a Reichert Ultracut-S microtome, picked up on to formvar/carbon-coated copper grids, sections were stained with 0.2% lead citrate. The sections were examined in a JEOL 1200EX transmission electron microscope and images were recorded with an AMT 2k CCD camera.

## Data visualization and statistics

All experiments were performed three times. For transmission, qPCR, FISH-based susceptibility assays, and lifespan assays data from each experiment are combined. For susceptibility assays and colonization assays a single representative experiment is shown and replicates are shown in the supplementary material. All qPCR-related data were analyzed in GraphPad Prism 10.2.3. All other data were analyzed in R Studio running R version 4.2.2 (*R Development Core Team, 2022*). When comparing all the means of more than two groups p-values were calculated using one-way ANOVA followed by the Tukey's HSD (honest significant difference) test. When comparing multiple experimental groups to a control group p-values were calculated using one-way ANOVA followed by Dunnett's test. p-Values for assays comparing only two groups were calculated using Student's t-test or Welch's t-test as indicated. *E. coli* OP50 is included in all experiments as a reference but was not included for statistical comparison unless explicitly noted. Susceptibility assay curves were modeled using the drc (*Ritz et al., 2015*) package in R. A two-parameter log-logistic function was used to model the curve and $ID_{50}$ values were calculated using the ED function. Kaplan-Meier survival curves were made using the survival (*Therneau, 2023*) package and curves were compared using the log-rank test within the survdiff function. Plots were made using the gdata (*Warnes, 2022*), scales (*Wickham and scales, 2022*), drc (*Ritz et al., 2015*), Rmisc (*Hope, 2022*), multcomp (*Hothorn et al., 2008*), survival (*Therneau, 2023*), ggplot2 (*Wickham, 2016*), ggsignif (*Ahlmann-Eltze and Patil, 2021*), and cowplot (*Wilke, 2020*) packages.

## Material availability statement

All reagents generated in this study (*Supplementary file 1a and b*) are available upon request. Primer sequence information is contained within *Supplementary file 2*.

## Acknowledgements

We would like to thank members of the Kim/Fischer laboratory for helpful comments during the preparation of this work. We would like to thank David Wang, Emily Troemel, Marie-Anne Félix, Eliana Drenkard, Simon Dove, E Peter Greenberg, Matthew Parsek, Jon Paczkowski, and Read Pukkila-Worley for kindly providing strains. Some strains were provided by the CGC, which is funded by NIH Office of Research Infrastructure Programs (P40 OD010440). We would like to further thank Brendan O'Hara, Michael Gebhardt, and Simon Dove for assistance in developing the *P. lurida* MYb11 transformation protocol. Electron microscopy consultation, sample fixing, and sample sectioning were performed in the HMS Electron Microscopy Facilty by Maria Ericsson and Anja Nordstrom. We additionally thank Rita Droste (MIT) for assistance with electron microscopy. BGV was partially supported by the MIT Department of Biology Graduate Program.

## Additional information

### Funding

| Funder | Grant reference number | Author |
|---|---|---|
| National Institutes of Health | R35GM141794 | Dennis H Kim |

The funders had no role in study design, data collection and interpretation, or the decision to submit the work for publication.

### Author contributions

Brian G Vassallo, Conceptualization, Formal analysis, Validation, Investigation, Visualization, Methodology, Writing – original draft, Project administration, Writing – review and editing; Noemie Scheidel, Conceptualization, Formal analysis, Validation, Investigation, Visualization, Methodology, Writing – review and editing; Sylvia E J Fischer, Conceptualization, Formal analysis, Supervision, Writing – review and editing; Dennis H Kim, Conceptualization, Formal analysis, Supervision, Funding acquisition, Writing – original draft, Writing – review and editing

### Author ORCIDs

Brian G Vassallo http://orcid.org/0000-0003-2645-664X
Noemie Scheidel http://orcid.org/0000-0002-7090-4123
Sylvia E J Fischer http://orcid.org/0000-0003-4290-0093
Dennis H Kim http://orcid.org/0000-0002-4109-5152

Reviewer #1 (Public review): https://doi.org/10.7554/eLife.92534.3.sa1
Reviewer #2 (Public review): https://doi.org/10.7554/eLife.92534.3.sa2
Author response https://doi.org/10.7554/eLife.92534.3.sa3

## Additional files

### Supplementary files

• Supplementary file 1. *C. elegans* strains, bacterial strains, and *Pseudomonas aeruginosa* PA14 transposon mutants used in this study.

• Supplementary file 2. Oligonucletides used in this study.

• Source data 1. Raw data included in this study.

• MDAR checklist

### Data availability

All raw data have been uploaded as *Source data 1*.

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
