## [Editor Report · eLife assessment]

Using a *C. elegans*/virus system, this **important** work demonstrates that viral susceptibility can be greatly altered by the bacterial food that *C. elegans* consumes. The work is rigorous with **solid** support for the conclusions: the authors show that quorum-sensing compounds play a role in reducing host susceptibility, and they perform control experiments to rule out nutrition and pathogenicity of the bacteria as the cause of impacts on viral susceptibility.

---

## [Referee Report · Reviewer #1 (Public review)]

Summary:

This manuscript explores the importance of food type on virus infection dynamics using a nematode virus as a model system. The authors demonstrate that susceptibility to viral infection can change by several orders of magnitude based on the type of bacterial food that potential hosts consume. They go on to show that, for the bacterial food source that reduces susceptibility, the effect is modulated by quorum sensing molecules that the bacteria produce.

Strengths:

This manuscript shows convincingly that nematode susceptibility to viral infection changes by several orders of magnitude (i.e. doses must be increased by several orders of magnitude to infect the same fraction of the population) depending on the bacterial food source on which hosts are reared. The authors then focus on the bacteria that reduce host susceptibility to viral infection and demonstrate that certain bacterial quorum-sensing compounds are required to see this effect of reduced susceptibility. Overall, sample sizes are large, methods are generally rigorous, experiments are repeated, and patterns are clear.

Comments on revised version:

The authors have now addressed all of my previous concerns.

---

## [Referee Report · Reviewer #2 (Public review)]

In this study, the authors investigate how diverse bacterial species influence Orsay virus transmission and host susceptibility in *C. elegans*. They find that Ochrobactrum species increase infection rates, while Pseudomonas species decrease infection rates, and they identify regulators of quorum sensing and the gacA two-component system as genetic factors in the effects of Pseudomonas on infection. These findings provide important insights into the species-specific effects that bacteria can have on viral infection in *C. elegans*, and they may have relevance for the impact of bacterial species on viral infection in other systems. Overall the manuscript has high rigor. However, a few minor concerns are listed below.

(1) The authors state that the amount of bacteria added to each plate was standardized by seeding plates with equivalent volumes of overnight culture. This approach does not account for differences in bacterial growth rate. A more rigorous approach would be to standardize based on OD600 measurements or CFU's. Alternatively, the authors could include bacterial growth curves to demonstrate that each strain/species has reached a similar growth phase (i.e. late log) at the time of plating, as bacterial physiology and virulence is dependent on the stage of growth. At the least, if it is not possible to perform these experiments, it would be useful to include a statement that potential differences in bacterial growth rate may influence their conclusions.

(2) Line 314-315: The claim "We did not observe any potent effect on host susceptibility to infection by Orsay virus from any supernatant (Supp. Fig. 9)" is not fully supported by the data, as the data in Fig S9 only show pals-5p::GFP levels. To confirm that host susceptibility is not affected, the authors would also measure the viral infection rate and/or viral load. Otherwise, the authors should rephrase the conclusion to increase accuracy. For example, "We did not observe any potent effect on pals-5p::GFP activation upon Orsay virus infection when animals were exposed to bacterial culture supernatant".

(3) The Ct values shown in Fig 3B-F should be normalized to a reference gene (i.e. Ct values for snb-1).

---

## [Author Response]

The following is the authors’ response to the original reviews.

**eLife assessment**
This important study identifies differential Orsay virus infection of *C. elegans* when animals are fed on different bacteria. The evidence for this is however, incomplete, as experiments to control for feeding rate and bacterial pathogenicity are needed as well as direct quantification of viral load.

We appreciate that the editors and reviewers felt that our manuscript addressed an important problem. We appreciate the constructive critiques provided by the reviewers and have worked to address all of the concerns, including a number of additional experiments as indicated below.

**Public Reviews:**

**Reviewer #1 (Public Review):**
Summary:This manuscript explores the importance of food type on virus infection dynamics using a nematode virus as a model system. The authors demonstrate that susceptibility to viral infection can change by several orders of magnitude based on the type of bacterial food that potential hosts consume. They go on to show that, for the bacterial food source that reduces susceptibility, the effect is modulated by quorum sensing molecules that the bacteria produce.Strengths:This manuscript shows convincingly that nematode susceptibility to viral infection changes by several orders of magnitude (i.e. doses must be increased by several orders of magnitude to infect the same fraction of the population) depending on the bacterial food source on which hosts are reared. The authors then focus on the bacteria that reduce host susceptibility to viral infection and demonstrate that certain bacterial quorum-sensing compounds are required to see this effect of reduced susceptibility. Overall, sample sizes are large, methods are generally rigorous, experiments are repeated, and patterns are clear.Weaknesses:Although the molecular correlate of reduced susceptibility is identified (i.e. quorum sensing compounds) the mechanisms underlying this effect are missing. For example, there are changes in susceptibility due to altered nutrition, host condition, the microbiome, feeding rate, mortality of infected hosts, etc. In addition, the authors focus almost entirely on the reduction in susceptibility even though I personally find the increased susceptibility generated when reared on Ochrobactrum to be much more exciting.I was a bit surprised that there was no data on basic factors that could have led to reductions in susceptibility. In particular, data on feeding rates and mortality rates seem really important. I would expect that feeding rates are reduced in the presence of Pseudomonas. Reduced feeding rates would translate to lower consumed doses, and so even though the same concentration of virus is on a plate, it doesn't mean that the same quantity of virus is consumed. Likewise, if Pseudomonas is causing mortality of virus-infected hosts, it could give the impression of lower infection rates. Perhaps mortality rates are too small in the experimental setup to explain this pattern, but that isn't clear in the current version of the manuscript. Is mortality greatly impacted by knocking out quorum-sensing genes? Also, the authors explored susceptibility to infection, but completely ignored variation in virus shedding.

We have added data on feeding rates (Line numbers 141-148 and 176-182, Supplementary Figure 4). After six hours of exposure no differences in feeding rate were observed. After 24 hours minor differences emerged between *O. vermis* MYb71 and each Pseudomonas species, however feeding rate inversely correlated with susceptibility to Orsay virus in that *O. vermis* MYb71 displayed the lowest feeding rate while *P. aeruginosa* PA14 displayed the highest feeding rate.

We have also added data on mortality rates (Line numbers 183-200, Supplementary Figure 6). No significant mortality was observed within the 24-hour exposure period used for our Orsay infection and transmission assays. *P. aeruginosa* virulence is dependent upon temperature and as our assays are done at 20°C rather than 25°C this may account for reduced mortality compared to other published results. Regardless, we noted that O. vermis MYb71 killed *C. elegans* as quickly as *P. aeruginosa* PA14 under these conditions and these two bacteria led to the shortest lifespan compared to the other tested bacteria. Interestingly, P. lurida MYb11 was observed to be more virulent than *P. aeruginosa* PA01 under these conditions. These results suggest that there is no direct correlation between mortality and susceptibility to Orsay virus, although it does not rule out that virulence effects unique to each bacterium could contribute to alterations in host susceptibility.

The reviewer is correct to assert that differences in viral shedding could exist. However, our susceptibility assays using exogenous Orsay virus remove this source of variation and yet we still observe the same trends such that O. vermis MYb71 promotes infection while P. lurida MYb11, *P. aeruginosa* PA01, and *P. aeruginosa* PA14 attenuate infection. Further we measured the amount of virus shed into the lawns in the presence of different bacteria and did not observe differences in shed virus that could account for the differences we observe in incidence proportion (Line numbers 241-254, Fig. 3 F). Viral stability could be an issue in both the transmission and susceptibility assays. We therefore tested viral stability in the presence of *E. coli*, P. lurida MYb11, *P. aeruginosa* PA01, and *P. aeruginosa* PA14 and successfully recovered virus from all lawns, suggesting virus is not rapidly degraded in the presence of any bacterium (Fig. 3D and 3E). However, we noted that the recovery of Orsay virus from lawns of *E. coli* OP50 and P. lurida MYb11 within 30 minutes was decreased compared to a spike-in control suggesting recovery from each lawn is not equivalent. This complicates a comparison of viral stability and shedding rates between different bacteria, but our ability to recover substantial amounts of virus in the shedding assay from the three Pseudomonas strains we examined precludes a substantial decrease in shedding rates as an explanation for the robust attenuation of Orsay virus observed in transmission assays.

I was also curious why the authors did not further explore the mechanism behind the quorumsensing effect. Not sure whether this is possible, but would it be possible to add spent media to the infection plates where the spent media was from Pseudomonas that produce the quorum sensing compound but the plates contain OP50, Pseudomonas, or the quorum sensing knockout of Pseudomonas? That would reveal whether it is the compound itself vs. something that the compound does.

We observed that quorum sensing mutants suppressed the attenuation of Orsay virus infection and we agree that this could be a consequence of the compounds themselves, or more likely an effect of the downstream consequences of quorum signaling. We added culture supernatant from each bacterium to lawns of *E. coli* OP50 to assess the effect on host susceptibility and did not observe any potent effect (Line numbers 311-318, Supplementary Figure 9). This supports an interpretation that it is not the compound itself that is responsible, however we cannot rule out that the compounds themselves may be responsible if provided at a higher concentration.

In addition, I was surprised by how much focus there was on the attenuation of infection and how little there was on the enhancement of infection. To me, enhancement seems like the more obvious thing to find a mechanism for -- is the bacteria suppressing immunity, preventing entry to gut cells, etc?

We are also intrigued by the enhancement of infection by Ochrobactrum spp, however we chose to focus on attenuation given the availability of *Pseudomonas aeruginosa* genetic mutants for study. We have added data (Line numbers 371-402, Figure 7, and Supplemental Figure 12) that inform our current hypothesis regarding Ochrobactrum mediated enhancement of Orsay virus infection.

I was a bit concerned about the "arbitrary units", which were used without any effort to normalize them. David Wang and Hongbing Jiang have developed a method based on tissue culture infectious dose 50 (TCID50) that can be used to measure infectious doses in a somewhat repeatable way. Without some type of normalization, it is hard to imagine how this study could be repeated. The 24-hour time period between exposure and glowing suggests very high doses, but it is still unclear precisely how high. Also, it is clear that multiple batches of virus were used in this study, but it is entirely unclear how variable these batches were.

We have clarified that we also measured the (TC)ID50 for every batch of virus used similar to the methods suggested by the Wang laboratory (Line numbers 107-119 and 499-506). We have added a figure showing the virus batch variability for all batches used in this study (Supp. Fig. 2). We have further clarified that the arbitrary units correspond to the actual microliters of viral filtrate used during infection and provided clear methods to replicate our viral batch production to assist with issues of reproducibility (Line numbers 107-119 and 499-506).

The authors in several places discuss high variability or low variability in incidence as though it is a feature of the virus or a feature of the host. It isn't. For infection data (or any type of binomial data) results are highly variable in the middle (close to 50% infection) and lowly variable at the ends (close to 0% or 100% infection). This is a result that is derived from a binomial distribution and it should not be taken as evidence that the bacteria or the host affect randomness. If you were to conduct dose-response experiments, on any of your bacterial food source treatments, you would find that variability is lowest at the extremely high and extremely low doses and it is most variable in the middle when you are at doses where about 50% of hosts are infected.

Thank you for pointing this out, we have removed all reference to this throughout the manuscript.

**Reviewer #2 (Public Review):**
Summary and Major Findings/Strengths:Across diverse hosts, microbiota can influence viral infection and transmission. *C. elegans* is naturally infected by the Orsay virus, which infects intestinal cells and is transmitted via the fecal-oral route. Previous work has demonstrated that host immune defense pathways, such as antiviral RNAi and the intracellular pathogen response (IPR), can influence host susceptibility to virus infection. However, little is known about how bacteria modulate viral transmission and host susceptibility.In this study, the authors investigate how diverse bacterial species influence Orsay virus transmission and host susceptibility in *C. elegans*. When *C. elegans* is grown in the presence of two Ochrobactrum species, the authors find that animals exhibit increased viral transmission, as measured by the increased proportion of newly infected worms (relative to growth on *E. coli* OP50). The presence of the two Ochrobactrum species also resulted in increased host susceptibility to the virus, which is reflected by the increased fraction of infected animals following exposure to the exogenous Orsay virus. In contrast, the presence of Pseudomonas lurida MYb11, as well as Pseudomonas PA01 or PA14, attenuates viral transmission and host susceptibility relative to *E. coli* OP50. For growth in the presence of *P. aeruginosa* PA01 and PA14, the attenuated transmission and susceptibility are suppressed by mutations in regulators of quorum sensing and the gacA two-component system. The authors also identify six virulence genes in *P. aeruginosa* PA14 that modulate host susceptibility to virus and viral transmission, albeit to a lesser extent. Based on the findings in *P. aeruginosa*, the authors further demonstrate that deletion of the gacA ortholog in *P. lurida* results in loss of the attenuation of viral transmission and host susceptibility.Taken together, these findings provide important insights into the species-specific effects that bacteria can have on viral infection in *C. elegans*. The authors also describe a role for Pseudomonas quorum sensing and virulence genes in influencing viral transmission and host susceptibility.Major weaknesses:The manuscript has several issues that need to be addressed, such as insufficient rigor of the experiments performed and questions about the reproducibility of the data presented in some places. In addition, confounding variables complicate the interpretations that can be made from the authors' findings and weaken some of the conclusions that are stated in the manuscript.(1) The authors sometimes use pals-5p::GFP expression to indicate infection, however, this is not necessarily an accurate measure of the infection rate. Specifically, in Figures 4-6, the authors should include measurements of viral RNA, either by FISH staining or qRT-PCR, to support the claims related to differences in infection rate.

Following the reviewers comment we have corroborated our pals-5::GFP data using FISH staining (Line numbers 291-292 and 357-359, Figure 4D & 4E, and Figure 6C).

(2) In several instances, the experimental setup and presentation of data lack sufficient rigor. For example, Fig 1D and Fig 2B only display data from one experimental replicate. The authors should include information from all 3 experimental replicates for more transparency. In Fig 3B, the authors should include a control that demonstrates how RNA1 levels change in the presence of *E. coli* OP50 for comparison with the results showing replication in the presence of PA14. In order to support the claim that "*P. aeruginosa* and P. lurida MYb11 do not eliminate Orsay virus infection", the authors should also measure RNA1 fold change in the presence of PA01 and P. lurida in the context of exogenous Orsay virus. Additionally, the authors should standardize the amount of bacteria added to the plate and specify how this was done in the Methods, as differing concentrations of bacteria could be the reason for species-specific effects on infection.

All experimental replicates are now included within the supplementary information.

We have also measured RNA1 fold change following infection in the presence of *P. aeruginosa* PA01 and P. lurida MYb11 (Line numbers Fig 3B and 3C) and found that these bacteria also do not eliminate Orsay virus replication.

We thank the reviewer for their comment on controlling the amount of bacteria and have clarified our methods section to more clearly explain that we seed our plates with equivalent amounts (based on volume) of overnight bacterial culture before allowing the bacteria to grow on the plates for 48 hours.

(3) The authors should be more careful about conclusions that are made from experiments involving PA14, which is a *P. aeruginosa* strain (isolated from humans), that can rapidly kill *C. elegans*. To eliminate confounding factors that are introduced by the pathogenicity of PA14, the authors should address how PA14 affects the health of the worms in their assays. For example, the authors should perform bead-feeding assays to demonstrate that feeding rates are unaffected when worms are grown in the presence of PA14. Because Orsay virus infection occurs through feeding, a decrease in *C. elegans* feeding rates can influence the outcome of viral infection. The authors should also address whether or not the presence of PA14 affects the stability of viral particles because that could be another trivial reason for the attenuation of viral infection that occurs in the presence of PA14.

We have added data on feeding rates (Line numbers 141-148 and 176-182, Supplementary Figure 4). After six hours of exposure no differences in feeding rate were observed. After 24 hours minor differences emerged between O. vermis MYb71 and each Pseudomonas species, however feeding rate inversely correlated with susceptibility to Orsay virus in that O. vermis MYb71 displayed the lowest feeding rate while *P. aeruginosa* PA14 displayed the highest feeding rate.

We have also added data on mortality rates (Line numbers 183-200, Supplementary Figure 6). No significant mortality was observed within the 24-hour exposure period used for our Orsay infection and transmission assays. *P. aeruginosa* virulence is dependent upon temperature and as our assays are done at 20°C rather than 25°C this may account for reduced mortality compared to other published results. Regardless, we noted that O. vermis MYb71 killed *C. elegans* as quickly as *P. aeruginosa* PA14 under these conditions and these two bacteria led to the shortest lifespan compared to the other tested bacteria. Interestingly, P. lurida MYb11 was observed to be more virulent than *P. aeruginosa* PA01 under these conditions. These results suggest that there is no direct correlation between mortality and susceptibility to Orsay virus, although it does not rule out that virulence effects unique to each bacterium could contribute to alterations in host susceptibility.

We tested viral stability in the presence of *E. coli* OP50 and Pseudomonas spp. and successfully recovered virus from all lawns, suggesting virus is not rapidly degraded in the presence of P. lurida MYb11, *P. aeruginosa* PA01, and *P. aeruginosa* PA14 (Line numbers 241-249, Fig 3D and Fig 3E). However, we noted that the recovery of Orsay virus from lawns of *E. coli* OP50 and P. lurida MYb11 within 30 minutes was decreased compared to a spike-in control suggesting recovery from each lawn is not equivalent. This complicates a comparison of viral stability and shedding rates between different bacteria, but our ability to recover substantial amounts of virus in the shedding assay from each Pseudomonas species precludes a substantial decrease in shedding rates as an explanation for the robust attenuation of Orsay virus observed in transmission assays.

**Recommendations for the authors:**

**Reviewer #1 (Recommendations For The Authors):**
Overall, I really liked this manuscript, I do think there are areas for improvement though.Some smaller things:Line 84: "can be observed spreading from a single animal" -- this isn't really great wording because the virus itself can't be observed (at least not very easily) -- even infection is hard to see.

The wording in line 84-85 has now been adjusted to read “can spread from a single animal”.

Fig 1C: which groups are statistically significantly different from each other?

Statistics have now been added to Figure 1C.

Line 154: not necessary to do for this paper, but this sentence made me curious whether the effect would have been seen with mixtures of bacteria (i.e. what if 50% were OP50 and 50% were Pseudomonas?)

This data has now been added in Line numbers 372-378, Figure 7A, and Supp. Fig. 12A and 12B.

Line 262-264: I don't find this interesting at all for the reasons mentioned earlier about binomial data being the most variable in the middle.

These lines have been removed.

Figure 4 B: The labels for the first two tick marks on the x-axis are switched I suspect. Otherwise, the controls did not behave as expected.

Figure 4B has been corrected.

Line 288, 297 and several other places: "Orsay Virus" should be "Orsay virus".

We have corrected these instances.

Supplemental Figure 2: Labels in the figure legend are B and C instead of A and B.

These labels have been adjusted for their placement within Figure 6.

Line 411: I suspect this was supposed to be 13,200 xg rather than 13.2 xg.

This error has been corrected.

Line 416-417: This sentence is very hard to interpret. More details are needed. This is the ID50 in which host strain? Is this averaged over all batches of virus? How variable are the batches?

This sentence (line number 114) has been amended to clarify that all ID50 values referred to here were calculated for ZD2611 populations in the presence of *E. coli* OP50. Further, Supplementary Figure 2 now shows all the ID50 values measured for each batch of virus used in this manuscript resulting in an average ID50 of 3.6.

Lines 467-469: Why exclude these instead of counting them as zeros in the analysis? How many plates fit this description -- were there lots or only a few over the course of all experiments?

We have chosen to exclude these plates as these samples lost spreaders at some point during the course of the assay potentially skewing the eventual number of new infections counted depending on when the infected spreader animal crawled off the plate. We have detailed the number of plates that fit this description in lines 559-562.

Line 476: A critical detail that is missing here is what number of worms were counted to score infection. Please say here or in the figure legends.

We have added the total number of worms counted and the minimum number counted per plate for each assay in the figure legends.

Line 546: Why was only a single representative experiment shown? I'm asking for a justification, not necessarily for you to show all the data.

We chose to show a single representative experiment for two reasons: We noted variability between susceptibility assays even when using the same batch of virus such that we could not combine experiments into a single plot as we did for transmission assays. Second, while we could normalize to a control within each experiment and expect to see similar relative differences across experiments, we believe this makes it more difficult to interpret the underlying data. For example, an increase in the infection rate of 80% compared to 10% within a population has only a single interpretation while a relative increase in the infection rate by 8x within a population could have several underlying meanings (e.g. 80% vs 10%, 64%vs 8%, 24% vs 3%). We have now included all experimental replicates in the supplementary material.

**Reviewer #2 (Recommendations For The Authors):**
Minor concerns:(1) Lines 86-87: "utilized a collection of bacteria isolated from the environment with wild *C. elegans*". The authors should provide more context on the source of these bacterial strains.

More references for the sources of these bacteria have been added to Supplementary Table 2.

(2) The presentation of data in Fig 1 could be improved. The authors should include the text "pals-5p::GFP" on the images shown in Fig 1B. The red dashed line in Fig. 1D should intersect the dose-response curve at y = 0.5. The column heading for Fig 1E states "ID50 +/- SD (a.u.)", but should read "ID50 ratio" and should not have units. It also might be more intuitive to normalize the ID50 value for O. vermis to *E. coli* OP50. This way, having an ID50 ratio >1 indicates decreased transmission relative to *E. coli*, and ID50 ratio <1 indicates increased transmission relative to *E. coli*. To increase the transparency and rigor of 1E, the authors should plot the ratios from all 3 experimental replicates. The authors should also briefly explain why different viral doses were used in Fig 1D and 1F.

The text “pals-5p::GFP” has now been added to Figure 1B and throughout the text. The red dashed line in figure 1D has been corrected. Figure 1E has been adjusted to an actual figure as suggested and the y-axis label is “ID50 Ratio Compared to *E. coli* OP50”. The ID50 replicates have been plotted in Supplementary Figure 2. We have clarified that the doses used are the same. Briefly, the technical replicates of individual doses from Figure 1D and Supplementary Figure 3A and 3B were pooled and processed for FISH staining to provide each experimental replicate of Figure 1F.

(3) Line 110: The claim is that Ochrobactrum and P. lurida MYb11 reduce the variability of infection levels. However, another possibility is that there's simply less dynamic range in the assay because the infection levels have been compressed to 100% and 0% under these conditions.

This line has been removed.

(4) There are discrepancies between what is shown in Fig 2C and what is described in the text. Lines 163-164: "*P. aeruginosa* PA01 and P. lurida MYb11 attenuated average infection to 33% and 62% of the population respectively". In Fig 2C, the mean for PA01 is ~25% whereas the mean for P. lurida appears to be less than 62%.

These values have been corrected.

(5) Line 196: Provide more context for why rde-1 mutants were tested. This is the first time rde-1 is mentioned in the text (i.e. why show results in rde-1 mutants when the results are in Fig 2).

More context has been provided for why rde-1 mutants were tested (Line numbers 228-232). Briefly, using the rde-1 mutant, which has defective antiviral immunity and therefore supports higher viral replication levels than the wild-type (Félix et al. 2011), allows us to potentiate our infection assay in Figure 3B and 3C such that we maximize our chances of detecting viral replication in the presence of the Pseudomonas species, and especially P. aeruginiosa PA14, where fewer animals might be expected to get infected based upon Figure 2B and Supplementary Figure 5.

(6) Lines 228-229: "Mutations of any the regulators of the las, rhl, or pqs quorum sensing systems suppressed the attenuation of Orsay virus infection caused by the presence of wild-type *P. aeruginosa* PA01". Based on this description, PA01 should have a lower fraction of GFP positive relative to the quorum sensing mutants in Fig 4B. It seems that the x-axis labels OP50 and PA01 are swapped.

The x-axis labels of Figure 4B have been corrected.

(7) To improve clarity, for any figures that have data showing the "fraction of individuals GFP positive", the authors should include "pals-5p::GFP" in the y-axis title and legend.

The y-axis labels, legends, and text have been corrected throughout.

(8) To improve overall clarity and flow, the order in which the data is presented could be reordered. In particular, Fig. 6 could be better positioned instead of being the last figure, as no further characterization is performed on the mutants, and the findings are not conserved in strains that are more relevant to the *C. elegans* microbiota, such as *P. lurida*. The overall story could be strengthened if the authors ended the manuscript with more details related to the mechanism by which regulators of quorum sensing modulate the outcome of viral infection.

Figure 5 and Figure 6 have now been swapped.

(9) Fig 5A: Make arrow sizes consistent across diagrams (i.e. the diagram for gacA deletion).

This figure (now Figure 6A) has been adjusted to make arrow sizes consistent across diagrams.

(10) Lines 280-282: "These data suggest that gacA has a conserved role across distant Pseudomonas species..." Here, the authors can provide more context on how well-conserved gacA is across Pseudomonas species (i.e. phylogenetic analysis of gacA sequences across different Pseudomonas species/strains). Furthermore, the data in Fig 5 does not provide strong enough support for the conclusion that gacA has a conserved role broadly across Pseudomonas species, as the authors only assess the effects of a gacA deletion in two species, *P. aeruginosa* and *P. lurida*.

We have adjusted lines 361-362 to “These data suggest that gacA has a conserved role between *P. aeruginosa* and P. lurida Myb11 in the attenuation of Orsay virus transmission and infection of *C. elegans*.” to reflect that we only assessed the effects of the gacA deletion in *P. aeruginosa* and *P. lurida* MYb11.

(11) The manuscript can be strengthened by performing additional experiments to elucidate the mechanism by which Pseudomonas modulates viral infection. Does the attenuation of viral transmission and host susceptibility by *P. lurida* and *P. aeruginosa* require *C. elegans* to be in the presence of live bacteria? For example, the authors could measure viral transmission and susceptibility of *C. elegans* grown on heat-killed Pseudomonas. Additionally, it would be interesting to determine if modulation of viral infection is dependent on a secreted molecule. To assess this, the authors could perform viral infections in the context of Pseudomonas culture supernatant.

We added bacterial culture supernatant from each bacterium to lawns of *E. coli* OP50 to assess the effect on host susceptibility and did not observe any potent effect (Line numbers 311-318, Supplementary Figure 9). This supports an interpretation that attenuation is not mediated by a secreted molecule, however we cannot rule out that attenuation activity would become apparent if supernatant were provided at a higher concentration.

We have found substantial challenges appropriately controlling live vs. heat-killed experiments particularly with the specifics of our susceptibility experiments. With regards to the underlying question of mechanism we believe that the genetic mutants (e.g. rhlR/gacA) are equally informative and that further comparison of these mutants’ interaction with the *C. elegans* host as compared to wild-type may be informative.

(12) The authors should include a discussion on the relative virulence potential of PA01, PA14, and P. lurida and the relationship between bacterial virulence potential and the outcome of viral infection.

We have also added data on mortality rates (Line numbers 183-200, Supplementary Figure 6). No significant mortality was observed within the 24-hour exposure period used for our Orsay infection and transmission assays. *P. aeruginosa* virulence is dependent upon temperature and as our assays are done at 20°C rather than 25°C this may account for reduced mortality compared to other published results. Regardless, we noted that O. vermis MYb71 killed *C. elegans* as quickly as *P. aeruginosa* PA14 under these conditions and these two bacteria led to the shortest lifespan compared to the other tested bacteria. Interestingly, P. lurida MYb11 was observed to be more virulent than *P. aeruginosa* PA01 under these conditions. These results suggest that there is no direct correlation between mortality and susceptibility to Orsay virus, although it does not rule out that virulence effects unique to each bacterium could contribute to alterations in host susceptibility.

(13) More information is needed on strains listed in Supplementary Table 2, particularly when there is no reference listed and the strain is "Gift of XXX lab". For example, the Troemel lab previously published about an Ochrobactrum strain in Troemel et al PLOS Biology 2008 PMID: 19071962 - is this the same strain? Please ensure that there is adequate information about each strain with as many published references as possible so that the work can be more easily reproduced.

We have added additional information and references to the strain table in Supplementary Table 2. The strain listed as Ochrobactrum sp. has been amended to Ochrobactrum BH3 as it is the strain described in Troemel et al. 2008.